# A Universal Primal-Dual Convex Optimization Framework

**Alp Yurtsever**[†]        **Quoc Tran-Dinh**[‡]        **Volkan Cevher**[†]

[†] Laboratory for Information and Inference Systems, EPFL, Switzerland
{alp.yurtsever, volkan.cevher}@epfl.ch
[‡] Department of Statistics and Operations Research, UNC, USA
quoctd@email.unc.edu

## Abstract

We propose a new primal-dual algorithmic framework for a prototypical constrained convex optimization template. The algorithmic instances of our framework are *universal* since they can automatically adapt to the unknown Hölder continuity degree and constant within the dual formulation. They are also guaranteed to have optimal convergence rates in the objective residual and the feasibility gap for each Hölder smoothness degree. In contrast to existing primal-dual algorithms, our framework avoids the proximity operator of the objective function. We instead leverage computationally cheaper, Fenchel-type operators, which are the main workhorses of the generalized conditional gradient (GCG)-type methods. In contrast to the GCG-type methods, our framework does not require the objective function to be differentiable, and can also process additional general linear inclusion constraints, while guarantees the convergence rate on the primal problem.

## 1  Introduction

This paper constructs an algorithmic framework for the following convex optimization template:

$$f^\star := \min_{\mathbf{x} \in \mathcal{X}} \{f(\mathbf{x}) : \mathbf{Ax} - \mathbf{b} \in \mathcal{K}\}, \qquad (1)$$

where $f : \mathbb{R}^p \to \mathbb{R} \cup \{+\infty\}$ is a convex function, $\mathbf{A} \in \mathbb{R}^{n \times p}$, $\mathbf{b} \in \mathbb{R}^n$, and $\mathcal{X}$ and $\mathcal{K}$ are nonempty, closed and convex sets in $\mathbb{R}^p$ and $\mathbb{R}^n$ respectively. The constrained optimization formulation (1) is quite flexible, capturing many important learning problems in a unified fashion, including matrix completion, sparse regularization, support vector machines, and submodular optimization [1–3].

Processing the inclusion $\mathbf{Ax} - \mathbf{b} \in \mathcal{K}$ in (1) requires a significant computational effort in the large-scale setting [4]. Hence, the majority of the scalable numerical solution methods for (1) are of the primal-dual-type, including decomposition, augmented Lagrangian, and alternating direction methods: *cf.*, [4–9]. The efficiency guarantees of these methods mainly depend on three properties of $f$: Lipschitz gradient, strong convexity, and the tractability of its proximal operator. For instance, the proximal operator of $f$, i.e., $\text{prox}_f(\mathbf{x}) := \arg\min_{\mathbf{z}} \{f(\mathbf{z}) + (1/2)\|\mathbf{z} - \mathbf{x}\|^2\}$, is key in handling non-smooth $f$ while obtaining the convergence rates as if it had Lipschitz gradient.

When the set $\mathbf{Ax} - \mathbf{b} \in \mathcal{K}$ is absent in (1), other methods can be preferable to primal-dual algorithms. For instance, if $f$ has Lipschitz gradient, then we can use the accelerated proximal gradient methods by applying the proximal operator for the indicator function of the set $\mathcal{X}$ [10, 11]. However, as the problem dimensions become increasingly larger, the proximal tractability assumption can be restrictive. This fact increased the popularity of the generalized conditional gradient (GCG) methods (or Frank-Wolfe-type algorithms), which instead leverage the following Fenchel-type oracles [1, 12, 13]

$$[\mathbf{x}]^\sharp_{\mathcal{X},g} := \arg\max_{\mathbf{s} \in \mathcal{X}} \{\langle \mathbf{x}, \mathbf{s} \rangle - g(\mathbf{s})\}, \qquad (2)$$

where $g$ is a convex function. When $g = 0$, we obtain the so-called linear minimization oracle [12]. When $\mathcal{X} \equiv \mathbb{R}^p$, then the (sub)gradient of the Fenchel conjugate of $g$, $\nabla g^*$, is in the set $[\mathbf{x}]^\sharp_g$.

The *sharp*-operator in (2) is often much cheaper to process as compared to the $\mathrm{prox}$ operator [1, 12]. While the GCG-type algorithms require $\mathcal{O}\left(1/\epsilon\right)$-iterations to guarantee an $\epsilon$ -primal objective residual/duality gap, they cannot converge when their objective is nonsmooth [14].

To this end, we propose a new primal-dual algorithmic framework that can exploit the sharp-operator of $f$ in lieu of its proximal operator. Our aim is to combine the flexibility of proximal primal-dual methods in addressing the general template (1) while leveraging the computational advantages of the GCG-type methods. As a result, we trade off the computational difficulty per iteration with the overall rate of convergence. While we obtain optimal rates based on the sharp-operator oracles, we note that the rates reduce to $\mathcal{O}\left(1/\epsilon^2\right)$ with the sharp operator vs. $\mathcal{O}\left(1/\epsilon\right)$ with the proximal operator when $f$ is completely non-smooth (*cf.* Definition 1.1). Intriguingly, the convergence rates are the same when $f$ is strongly convex. Unlike GCG-type methods, our approach can now handle nonsmooth objectives in addition to complex constraint structures as in (1).

Our primal-dual framework is *universal* in the sense the convergence of our algorithms can optimally adapt to the Hölder continuity of the dual objective $g$ (*cf.*, (6) in Section 3) without having to know its parameters. By Hölder continuity, we mean the (sub)gradient $\nabla g$ of a convex function $g$ satisfies $\|\nabla g(\boldsymbol{\lambda}) - \nabla g(\tilde{\boldsymbol{\lambda}})\| \leqslant M_\nu \|\boldsymbol{\lambda} - \tilde{\boldsymbol{\lambda}}\|^\nu$ with parameters $M_\nu < \infty$ and $\nu \in [0,1]$ for all $\boldsymbol{\lambda}, \tilde{\boldsymbol{\lambda}} \in \mathbb{R}^n$. The case $\nu = 0$ models the bounded subgradient, whereas $\nu = 1$ captures the Lipschitz gradient. The Hölder continuity has recently resurfaced in unconstrained optimization by [15] with universal gradient methods that obtain optimal rates without having to know $M_\nu$ and $\nu$. Unfortunately, these methods cannot directly handle the general constrained template (1). After our initial draft appeared, [14] presented new GCG-type methods for composite minimization, i.e., $\min_{\mathbf{x}\in\mathbb{R}^p} f(\mathbf{x}) + \psi(\mathbf{x})$, relying on Hölder *smoothness* of $f$ (i.e., $\nu \in (0,1]$) and the sharp-operator of $\psi$. The methods in [14] do not apply when $f$ is non-smooth. In addition, they cannot process the additional inclusion $\mathbf{Ax} - \mathbf{b} \in \mathcal{K}$ in (1), which is a major drawback for machine learning applications.

Our algorithmic framework features a gradient method and its accelerated variant that operates on the dual formulation of (1). For the accelerated variant, we study an alternative to the universal accelerated method of [15] based on FISTA [10] since it requires less proximal operators in the dual. While the FISTA scheme is classical, our analysis of it with the Hölder continuous assumption is new. Given the dual iterates, we then use a new averaging scheme to construct the primal-iterates for the constrained template (1). In contrast to the non-adaptive weighting schemes of GCG-type algorithms, our weights explicitly depend on the local estimates of the Hölder constants $M_\nu$ at each iteration. Finally, we derive the worst-case complexity results. Our results are optimal since they match the computational lowerbounds in the sense of first-order black-box methods [16].

**Paper organization:** Section 2 briefly recalls primal-dual formulation of problem (1) with some standard assumptions. Section 3 defines the universal gradient mapping and its properties. Section 4 presents the primal-dual universal gradient methods (both the standard and accelerated variants), and analyzes their convergence. Section 5 provides numerical illustrations, followed by our conclusions. The supplementary material includes the technical proofs and additional implementation details.

**Notation and terminology:** For notational simplicity, we work on the $\mathbb{R}^p/\mathbb{R}^n$ spaces with the Euclidean norms. We denote the Euclidean distance of the vector $\mathbf{u}$ to a closed convex set $\mathcal{X}$ by $\mathrm{dist}\left(\mathbf{u}, \mathcal{X}\right)$. Throughout the paper, $\|\cdot\|$ represents the Euclidean norm for vectors and the spectral norm for the matrices. For a convex function $f$, we use $\nabla f$ both for its subgradient and gradient, and $f^*$ for its Fenchel's conjugate. Our goal is to approximately solve (1) to obtain $\mathbf{x}_\epsilon$ in the following sense:

**Definition 1.1.** *Given an accuracy level $\epsilon > 0$, a point $\mathbf{x}_\epsilon \in \mathcal{X}$ is said to be an $\epsilon$-solution of* (1) *if*

$$|f(\mathbf{x}_\epsilon) - f^\star| \leqslant \epsilon, \ \ and \ \ \mathrm{dist}\left(\mathbf{Ax}_\epsilon - \mathbf{b}, \mathcal{K}\right) \leqslant \epsilon.$$

Here, we call $|f(\mathbf{x}_\epsilon) - f^\star|$ the primal objective residual and $\mathrm{dist}\left(\mathbf{Ax}_\epsilon - \mathbf{b}, \mathcal{K}\right)$ the feasibility gap.

## 2 Primal-dual preliminaries

In this section, we briefly summarize the primal-dual formulation with some standard assumptions. For the ease of presentation, we reformulate (1) by introducing a slack variable $\mathbf{r}$ as follows:

$$f^\star = \min_{\mathbf{x}\in\mathcal{X}, \mathbf{r}\in\mathcal{K}} \{f(\mathbf{x}) : \mathbf{Ax} - \mathbf{r} = \mathbf{b}\}, \ (\mathbf{x}^\star : f(\mathbf{x}^\star) = f^\star). \tag{3}$$

Let $\mathbf{z} := [\mathbf{x}, \mathbf{r}]$ and $\mathcal{Z} := \mathcal{X} \times \mathcal{K}$. Then, we have $\mathcal{D} := \{\mathbf{z} \in \mathcal{Z} : \mathbf{Ax} - \mathbf{r} = \mathbf{b}\}$ as the feasible set of (3).

**The dual problem:** The Lagrange function associated with the linear constraint $\mathbf{Ax} - \mathbf{r} = \mathbf{b}$ is defined as $\mathcal{L}(\mathbf{x}, \mathbf{r}, \boldsymbol{\lambda}) := f(\mathbf{x}) + \langle \boldsymbol{\lambda}, \mathbf{Ax} - \mathbf{r} - \mathbf{b} \rangle$, and the dual function $d$ of (3) can be defined and decomposed as follows:

$$d(\boldsymbol{\lambda}) := \min_{\substack{\mathbf{x} \in \mathcal{X} \\ \mathbf{r} \in \mathcal{K}}} \{f(\mathbf{x}) + \langle \boldsymbol{\lambda}, \mathbf{Ax} - \mathbf{r} - \mathbf{b} \rangle\} = \underbrace{\min_{\mathbf{x} \in \mathcal{X}} \{f(\mathbf{x}) + \langle \boldsymbol{\lambda}, \mathbf{Ax} - \mathbf{b} \rangle\}}_{d_x(\boldsymbol{\lambda})} + \underbrace{\min_{\mathbf{r} \in \mathcal{K}} \langle \boldsymbol{\lambda}, -\mathbf{r} \rangle}_{d_r(\boldsymbol{\lambda})},$$

where $\boldsymbol{\lambda} \in \mathbb{R}^n$ is the dual variable. Then, we define the dual problem of (3) as follows:

$$d^\star := \max_{\boldsymbol{\lambda} \in \mathbb{R}^n} d(\boldsymbol{\lambda}) = \max_{\boldsymbol{\lambda} \in \mathbb{R}^n} \left\{ d_x(\boldsymbol{\lambda}) + d_r(\boldsymbol{\lambda}) \right\}. \tag{4}$$

**Fundamental assumptions:** To characterize the primal-dual relation between (1) and (4), we require the following assumptions [17]:

*Assumption A.* 1. The function $f$ is proper, closed, and convex, but not necessarily smooth. The constraint sets $\mathcal{X}$ and $\mathcal{K}$ are nonempty, closed, and convex. The solution set $\mathcal{X}^\star$ of (1) is nonempty. Either $\mathcal{Z}$ is polyhedral or the *Slater's condition* holds. By the Slater's condition, we mean $\mathrm{ri}(\mathcal{Z}) \cap \{(\mathbf{x}, \mathbf{r}) : \mathbf{Ax} - \mathbf{r} = \mathbf{b}\} \neq \varnothing$, where $\mathrm{ri}(\mathcal{Z})$ stands for the relative interior of $\mathcal{Z}$.

**Strong duality:** Under Assumption A.1, the solution set $\boldsymbol{\Lambda}^\star$ of the dual problem (4) is also nonempty and bounded. Moreover, the *strong duality* holds, i.e., $f^\star = d^\star$.

# 3 Universal gradient mappings

This section defines the universal gradient mapping and its properties.

## 3.1 Dual reformulation

We first adopt the composite convex minimization formulation of (4) for better interpretability as

$$G^\star := \min_{\boldsymbol{\lambda} \in \mathbb{R}^n} \{G(\boldsymbol{\lambda}) := g(\boldsymbol{\lambda}) + h(\boldsymbol{\lambda})\}, \tag{5}$$

where $G^\star = -d^\star$, and the correspondence between $(g, h)$ and $(d_x, d_r)$ is as follows:

$$\begin{cases} g(\boldsymbol{\lambda}) & := \max_{\mathbf{x} \in \mathcal{X}} \{\langle \boldsymbol{\lambda}, \mathbf{b} - \mathbf{Ax} \rangle - f(\mathbf{x})\} = -d_x(\boldsymbol{\lambda}), \\ h(\boldsymbol{\lambda}) & := \max_{\mathbf{r} \in \mathcal{K}} \langle \boldsymbol{\lambda}, \mathbf{r} \rangle = -d_r(\boldsymbol{\lambda}). \end{cases} \tag{6}$$

Since $g$ and $h$ are generally non-smooth, FISTA and its proximal-based analysis [10] are not directly applicable. Recall the sharp operator defined in (2), then $g$ can be expressed as

$$g(\boldsymbol{\lambda}) = \max_{\mathbf{x} \in \mathcal{X}} \left\{ \langle -\mathbf{A}^T \boldsymbol{\lambda}, \mathbf{x} \rangle - f(\mathbf{x}) \right\} + \langle \boldsymbol{\lambda}, \mathbf{b} \rangle,$$

and we define the optimal solution to the $g$ subproblem above as follows:

$$\mathbf{x}^*(\boldsymbol{\lambda}) \in \arg\max_{\mathbf{x} \in \mathcal{X}} \left\{ \langle -\mathbf{A}^T \boldsymbol{\lambda}, \mathbf{x} \rangle - f(\mathbf{x}) \right\} \equiv [-\mathbf{A}^T \boldsymbol{\lambda}]^\sharp_{\mathcal{X}, f}. \tag{7}$$

The second term, $h$, depends on the structure of $\mathcal{K}$. We consider three special cases:

(a) **Sparsity/low-rankness:** If $\mathcal{K} := \{\mathbf{r} \in \mathbb{R}^n : \|\mathbf{r}\| \leqslant \kappa\}$ for a given $\kappa \geqslant 0$ and a given norm $\|\cdot\|$, then $h(\boldsymbol{\lambda}) = \kappa \|\boldsymbol{\lambda}\|^*$, the scaled dual norm of $\|\cdot\|$. For instance, if $\mathcal{K} := \{\mathbf{r} \in \mathbb{R}^n : \|\mathbf{r}\|_1 \leqslant \kappa\}$, then $h(\boldsymbol{\lambda}) = \kappa \|\boldsymbol{\lambda}\|_\infty$. While the $\ell_1$-norm induces the sparsity of $\mathbf{x}$, computing $h$ requires the max absolute elements of $\boldsymbol{\lambda}$. If $\mathcal{K} := \{\mathbf{r} \in \mathbb{R}^{q_1 \times q_2} : \|\mathbf{r}\|_* \leqslant \kappa\}$ (the nuclear norm), then $h(\boldsymbol{\lambda}) = \kappa \|\boldsymbol{\lambda}\|$, the spectral norm. The nuclear norm induces the low-rankness of $\mathbf{x}$. Computing $h$ in this case leads to finding the top-eigenvalue of $\boldsymbol{\lambda}$, which is efficient.

(b) **Cone constraints:** If $\mathcal{K}$ is a cone, then $h$ becomes the indicator function $\delta_{\mathcal{K}*}$ of its dual cone $\mathcal{K}^*$. Hence, we can handle the inequality constraints and positive semidefinite constraints in (1). For instance, if $\mathcal{K} \equiv \mathbb{R}^n_+$, then $h(\boldsymbol{\lambda}) = \delta_{\mathbb{R}^n_-}(\boldsymbol{\lambda})$, the indicator function of $\mathbb{R}^n_- := \{\boldsymbol{\lambda} \in \mathbb{R}^n : \boldsymbol{\lambda} \leqslant 0\}$. If $\mathcal{K} \equiv \mathcal{S}^p_+$, then $h(\boldsymbol{\lambda}) := \delta_{\mathcal{S}^p_-}(\boldsymbol{\lambda})$, the indicator function of the negative semidefinite matrix cone.

(c) **Separable structures:** If $\mathcal{X}$ and $f$ are separable, i.e., $\mathcal{X} := \prod_{i=1}^p \mathcal{X}_i$ and $f(\mathbf{x}) := \sum_{i=1}^p f_i(\mathbf{x}_i)$, then the evaluation of $g$ and its derivatives can be decomposed into $p$ subproblems.

### 3.2 Hölder continuity of the dual universal gradient

Let $\nabla g(\cdot)$ be a subgradient of $g$, which can be computed as $\nabla g(\boldsymbol{\lambda}) = \mathbf{b} - \mathbf{A}\mathbf{x}^*(\boldsymbol{\lambda})$. Next, we define

$$M_\nu = M_\nu(g) := \sup_{\boldsymbol{\lambda}, \tilde{\boldsymbol{\lambda}} \in \mathbb{R}^n, \boldsymbol{\lambda} \neq \tilde{\boldsymbol{\lambda}}} \left\{ \frac{\|\nabla g(\boldsymbol{\lambda}) - \nabla g(\tilde{\boldsymbol{\lambda}})\|}{\|\boldsymbol{\lambda} - \tilde{\boldsymbol{\lambda}}\|^\nu} \right\}, \tag{8}$$

where $\nu \geqslant 0$ is the Hölder smoothness order. Note that the parameter $M_\nu$ explicitly depends on $\nu$ [15]. We are interested in the case $\nu \in [0, 1]$, and *especially the two extremal cases*, where we either have the Lipschitz gradient that corresponds to $\nu = 1$, or the bounded subgradient that corresponds to $\nu = 0$.

We require the following condition in the sequel:

*Assumption A. 2.* $\hat{M}(g) := \inf_{0 \leqslant \nu \leqslant 1} M_\nu(g) < +\infty$.

Assumption A.2 is reasonable. We explain this claim with the following two examples. First, if $g$ is subdifferentiable and $\mathcal{X}$ is bounded, then $\nabla g(\cdot)$ is also bounded. Indeed, we have

$$\|\nabla g(\boldsymbol{\lambda})\| = \|\mathbf{b} - \mathbf{A}\mathbf{x}^*(\boldsymbol{\lambda})\| \leqslant D_\mathcal{X}^\mathbf{A} := \sup\{\|\mathbf{b} - \mathbf{A}\mathbf{x}\| : \mathbf{x} \in \mathcal{X}\}.$$

Hence, we can choose $\nu = 0$ and $\hat{M}_\nu(g) = 2D_\mathcal{X}^\mathbf{A} < \infty$.

Second, if $f$ is uniformly convex with the convexity parameter $\mu_f > 0$ and the degree $q \geqslant 2$, i.e., $\langle \nabla f(\mathbf{x}) - \nabla f(\tilde{\mathbf{x}}), \mathbf{x} - \tilde{\mathbf{x}} \rangle \geqslant \mu_f \|\mathbf{x} - \tilde{\mathbf{x}}\|^q$ for all $\mathbf{x}, \tilde{\mathbf{x}} \in \mathbb{R}^p$, then $g$ defined by (6) satisfies (8) with $\nu = \frac{1}{q-1}$ and $\hat{M}_\nu(g) = \left(\mu_f^{-1}\|\mathbf{A}\|^2\right)^{\frac{1}{q-1}} < +\infty$, as shown in [15]. In particular, if $q = 2$, i.e., $f$ is $\mu_f$-strongly convex, then $\nu = 1$ and $M_\nu(g) = \mu_f^{-1}\|\mathbf{A}\|^2$, which is the Lipschitz constant of the gradient $\nabla g$.

### 3.3 The proximal-gradient step for the dual problem

Given $\hat{\boldsymbol{\lambda}}_k \in \mathbb{R}^n$ and $M_k > 0$, we define

$$Q_{M_k}(\boldsymbol{\lambda}; \hat{\boldsymbol{\lambda}}_k) := g(\hat{\boldsymbol{\lambda}}_k) + \langle \nabla g(\hat{\boldsymbol{\lambda}}_k), \boldsymbol{\lambda} - \hat{\boldsymbol{\lambda}}_k \rangle + \frac{M_k}{2}\|\boldsymbol{\lambda} - \hat{\boldsymbol{\lambda}}_k\|^2$$

as an approximate quadratic surrogate of $g$. Then, we consider the following update rule:

$$\boldsymbol{\lambda}_{k+1} := \arg\min_{\boldsymbol{\lambda} \in \mathbb{R}^n} \left\{ Q_{M_k}(\boldsymbol{\lambda}; \hat{\boldsymbol{\lambda}}_k) + h(\boldsymbol{\lambda}) \right\} \equiv \mathrm{prox}_{M_k^{-1}h}\left(\hat{\boldsymbol{\lambda}}_k - M_k^{-1}\nabla g(\hat{\boldsymbol{\lambda}}_k)\right). \tag{9}$$

For a given accuracy $\epsilon > 0$, we define

$$\overline{M}_\epsilon := \left[\frac{1-\nu}{1+\nu}\frac{1}{\epsilon}\right]^{\frac{1-\nu}{1+\nu}} M_\nu^{\frac{2}{1+\nu}}. \tag{10}$$

We need to choose the parameter $M_k > 0$ such that $Q_{M_k}$ is an approximate upper surrogate of $g$, i.e., $g(\boldsymbol{\lambda}) \leqslant Q_{M_k}(\boldsymbol{\lambda}; \boldsymbol{\lambda}_k) + \delta_k$ for some $\boldsymbol{\lambda} \in \mathbb{R}^n$ and $\delta_k \geqslant 0$. If $\nu$ and $M_\nu$ are known, then we can set $M_k = \overline{M}_\epsilon$ defined by (10). In this case, $Q_{\overline{M}_\epsilon}$ is an upper surrogate of $g$. In general, we do not know $\nu$ and $M_\nu$. Hence, $M_k$ can be determined via a backtracking line-search procedure.

## 4 Universal primal-dual gradient methods

We apply the universal gradient mappings to the dual problem (5), and propose an averaging scheme to construct $\{\bar{\mathbf{x}}_k\}$ for approximating $\mathbf{x}^\star$. Then, we develop an accelerated variant based on the FISTA scheme [10], and construct another primal sequence $\{\bar{\bar{\mathbf{x}}}_k\}$ for approximating $\mathbf{x}^\star$.

### 4.1 Universal primal-dual gradient algorithm

Our algorithm is shown in Algorithm 1. The dual steps are simply the universal gradient method in [15], while the new primal step allows to approximate the solution of (1).

*Complexity-per-iteration:* First, computing $\mathbf{x}^*(\boldsymbol{\lambda}_k)$ at Step 1 requires the solution $\mathbf{x}^*(\boldsymbol{\lambda}_k) \in [-\mathbf{A}^T\boldsymbol{\lambda}_k]_{\mathcal{X}, f}^\sharp$. For many $\mathcal{X}$ and $f$, we can compute $\mathbf{x}^*(\boldsymbol{\lambda}_k)$ efficiently and often in a closed form.

---

**Algorithm 1** (*Universal Primal-Dual Gradient Method* (UniPDGrad))

---

**Initialization:** Choose an initial point $\boldsymbol{\lambda}_0 \in \mathbb{R}^n$ and a desired accuracy level $\epsilon > 0$.
Estimate a value $M_{-1}$ such that $0 < M_{-1} \leqslant \overline{M}_\epsilon$. Set $S_{-1} = 0$ and $\bar{\mathbf{x}}_{-1} = \mathbf{0}^p$.
**for** $k = 0$ **to** $k_{\max}$
   1. Compute a primal solution $\mathbf{x}^*(\boldsymbol{\lambda}_k) \in [-\mathbf{A}^T \boldsymbol{\lambda}_k]_{\mathcal{X},f}^\sharp$.
   2. Form $\nabla g(\boldsymbol{\lambda}_k) = \mathbf{b} - \mathbf{A}\mathbf{x}^*(\boldsymbol{\lambda}_k)$.
   3. **Line-search:** Set $M_{k,0} = 0.5 M_{k-1}$. For $i = 0$ to $i_{\max}$, perform the following steps:
     3.a. Compute the trial point $\boldsymbol{\lambda}_{k,i} = \text{prox}_{M_{k,i}^{-1} h}\left(\boldsymbol{\lambda}_k - M_{k,i}^{-1} \nabla g(\boldsymbol{\lambda}_k)\right)$.
     3.b. If the following line-search condition holds:

$$g(\boldsymbol{\lambda}_{k,i}) \leqslant Q_{M_{k,i}}(\boldsymbol{\lambda}_{k,i}; \boldsymbol{\lambda}_k) + \epsilon/2,$$

       then set $i_k = i$ and terminate the line-search loop. Otherwise, set $M_{k,i+1} = 2 M_{k,i}$.
   **End of line-search**
   4. Set $\boldsymbol{\lambda}_{k+1} = \boldsymbol{\lambda}_{k,i_k}$ and $M_k = M_{k,i_k}$. Compute $w_k = \frac{1}{M_k}$, $S_k = S_{k-1} + w_k$, and $\gamma_k = \frac{w_k}{S_k}$.
   5. Compute $\bar{\mathbf{x}}_k = (1 - \gamma_k)\bar{\mathbf{x}}_{k-1} + \gamma_k \mathbf{x}^*(\boldsymbol{\lambda}_k)$.
**end for**
**Output:** Return the primal approximation $\bar{\mathbf{x}}_k$ for $\mathbf{x}^\star$.

---

Second, in the line-search procedure, we require the solution $\boldsymbol{\lambda}_{k,i}$ at Step 3.a, and the evaluation of $g(\boldsymbol{\lambda}_{k,i})$. The total computational cost depends on the proximal operator of $h$ and the evaluations of $g$. We prove below that our algorithm requires two oracle queries of $g$ on average.

**Theorem 4.1.** *The primal sequence* $\{\bar{\mathbf{x}}_k\}$ *generated by the Algorithm 1 satisfies*

$$-\|\boldsymbol{\lambda}^\star\| \text{dist}(\mathbf{A}\bar{\mathbf{x}}_k - \mathbf{b}, \mathcal{K}) \leqslant f(\bar{\mathbf{x}}^k) - f^\star \leqslant \frac{\overline{M}_\epsilon \|\boldsymbol{\lambda}_0\|^2}{k+1} + \frac{\epsilon}{2}, \tag{11}$$

$$\text{dist}(\mathbf{A}\bar{\mathbf{x}}_k - \mathbf{b}, \mathcal{K}) \leqslant \frac{4\overline{M}_\epsilon}{k+1}\|\boldsymbol{\lambda}_0 - \boldsymbol{\lambda}^\star\| + \sqrt{\frac{2\overline{M}_\epsilon \epsilon}{k+1}}, \tag{12}$$

*where* $\overline{M}_\epsilon$ *is defined by* (10), $\boldsymbol{\lambda}^\star \in \Lambda^\star$ *is an arbitrary dual solution, and* $\epsilon$ *is the desired accuracy.*

*The worst-case analytical complexity:* We establish the total number of iterations $k_{\max}$ to achieve an $\epsilon$-solution $\bar{\mathbf{x}}_k$ of (1). The supplementary material proves that

$$k_{\max} = \left\lfloor \left[\frac{4\sqrt{2}\|\boldsymbol{\lambda}^\star\|}{-1 + \sqrt{1 + 8\frac{\|\boldsymbol{\lambda}^\star\|}{\|\boldsymbol{\lambda}^\star\|_{[1]}}}}\right]^2 \inf_{0 \leqslant \nu \leqslant 1} \left(\frac{M_\nu}{\epsilon}\right)^{\frac{2}{1+\nu}} \right\rfloor, \tag{13}$$

where $\|\boldsymbol{\lambda}^\star\|_{[1]} = \max\{\|\boldsymbol{\lambda}^\star\|, 1\}$. This complexity is optimal for $\nu = 0$, but not for $\nu > 0$ [16].

At each iteration $k$, the linesearch procedure at Step 3 requires the evaluations of $g$. The supplementary material bounds the total number $N_1(k)$ of oracle queries, including the function $G$ and its gradient evaluations, up to the $k$th iteration as follows:

$$N_1(k) \leqslant 2(k+1) + 1 - \log_2(M_{-1}) + \inf_{0 \leqslant \nu \leqslant 1} \left\{\frac{1-\nu}{1+\nu}\log_2\left(\frac{(1-\nu)}{(1+\nu)\epsilon}\right) + \frac{2}{1+\nu}\log_2 M_\nu\right\}. \tag{14}$$

Hence, we have $N_1(k) \approx 2(k+1)$, i.e., we require approximately two oracle queries at each iteration on the average.

## 4.2 Accelerated universal primal-dual gradient method

We now develop an accelerated scheme for solving (5). Our scheme is different from [15] in two key aspects. First, we adopt the FISTA [10] scheme to obtain the dual sequence since it requires less prox operators compared to the fast scheme in [15]. Second, we perform the line-search after computing $\nabla g(\hat{\boldsymbol{\lambda}}_k)$, which can reduce the number of the sharp-operator computations of $f$ and $\mathcal{X}$. Note that the application of FISTA to the dual function is not novel per se. However, we claim that our theoretical characterization of this classical scheme based on the Hölder continuity assumption in the composite minimization setting is new.

---

**Algorithm 2** (*Accelerated Universal Primal-Dual Gradient Method* (AccUniPDGrad))

---

**Initialization:** Choose an initial point $\boldsymbol{\lambda}_0 = \hat{\boldsymbol{\lambda}}_0 \in \mathbb{R}^n$ and an accuracy level $\epsilon > 0$.
Estimate a value $M_{-1}$ such that $0 < M_{-1} \leqslant \overline{M}_\epsilon$. Set $\hat{S}_{-1} = 0$, $t_0 = 1$ and $\bar{\bar{\mathbf{x}}}_{-1} = \mathbf{0}^p$.
**for** $k = 0$ **to** $k_{\max}$
    1. Compute a primal solution $\mathbf{x}^*(\hat{\boldsymbol{\lambda}}_k) \in [-\mathbf{A}^T\hat{\boldsymbol{\lambda}}]_{\mathcal{X},f}^\sharp$.
    2. Form $\nabla g(\hat{\boldsymbol{\lambda}}_k) = \mathbf{b} - \mathbf{A}\mathbf{x}^*(\hat{\boldsymbol{\lambda}}_k)$.
    3. **Line-search:** Set $M_{k,0} = M_{k-1}$. For $i = 0$ to $i_{\max}$, perform the following steps:
      3.a. Compute the trial point $\boldsymbol{\lambda}_{k,i} = \text{prox}_{M_{k,i}^{-1}h}\big(\hat{\boldsymbol{\lambda}}_k - M_{k,i}^{-1}\nabla g(\hat{\boldsymbol{\lambda}}_k)\big)$.
      3.b. If the following line-search condition holds:

$$g(\boldsymbol{\lambda}_{k,i}) \leqslant Q_{M_{k,i}}(\boldsymbol{\lambda}_{k,i}; \hat{\boldsymbol{\lambda}}_k) + \epsilon/(2t_k),$$

      then $i_k = i$, and terminate the line-search loop. Otherwise, set $M_{k,i+1} = 2M_{k,i}$.
    **End of line-search**
    4. Set $\boldsymbol{\lambda}_{k+1} = \boldsymbol{\lambda}_{k,i_k}$ and $M_k = M_{k,i_k}$. Compute $w_k = \frac{t_k}{M_k}$, $\hat{S}_k = \hat{S}_{k-1} + w_k$, and $\gamma_k = w_k/\hat{S}_k$.
    5. Compute $t_{k+1} = 0.5\big[1 + \sqrt{1 + 4t_k^2}\big]$ and update $\hat{\boldsymbol{\lambda}}_{k+1} = \boldsymbol{\lambda}_{k+1} + \frac{t_k-1}{t_{k+1}}\big(\boldsymbol{\lambda}_{k+1} - \boldsymbol{\lambda}_k\big)$.
    6. Compute $\bar{\bar{\mathbf{x}}}_k = (1 - \gamma_k)\bar{\bar{\mathbf{x}}}_{k-1} + \gamma_k\mathbf{x}^*(\hat{\boldsymbol{\lambda}}_k)$.
**end for**
**Output:** Return the primal approximation $\bar{\bar{\mathbf{x}}}_k$ for $\mathbf{x}^\star$.

---

*Complexity per-iteration:* The per-iteration complexity of Algorithm 2 remains essentially the same as that of Algorithm 1.

**Theorem 4.2.** *The primal sequence $\{\bar{\bar{\mathbf{x}}}_k\}$ generated by the Algorithm 2 satisfies*

$$-\|\boldsymbol{\lambda}^\star\|\text{dist}\,(\mathbf{A}\bar{\bar{\mathbf{x}}}_k - \mathbf{b}, \mathcal{K}) \leqslant f(\bar{\mathbf{x}}^k) - f^\star \leqslant \frac{\epsilon}{2} + \frac{4\overline{M}_\epsilon\|\boldsymbol{\lambda}_0\|^2}{(k+2)^{\frac{1+3\nu}{1+\nu}}}, \tag{15}$$

$$\text{dist}\,(\mathbf{A}\bar{\bar{\mathbf{x}}}_k - \mathbf{b}, \mathcal{K}) \leqslant \frac{16\overline{M}_\epsilon}{(k+2)^{\frac{1+3\nu}{1+\nu}}}\|\boldsymbol{\lambda}_0 - \boldsymbol{\lambda}^\star\| + \sqrt{\frac{8\overline{M}_\epsilon\epsilon}{(k+2)^{\frac{1+3\nu}{1+\nu}}}}, \tag{16}$$

*where $\overline{M}_\epsilon$ is defined by* (10), *$\boldsymbol{\lambda}^\star \in \boldsymbol{\Lambda}^\star$ is an arbitrary dual solution, and $\epsilon$ is the desired accuracy.*

*The worst-case analytical complexity:* The supplementary material proves the following worst-case complexity of Algorithm 2 to achieve an $\epsilon$-solution $\bar{\bar{\mathbf{x}}}_k$:

$$k_{\max} = \left\lfloor \left[\frac{8\sqrt{2}\|\boldsymbol{\lambda}^\star\|}{-1 + \sqrt{1 + 8\frac{\|\boldsymbol{\lambda}\|}{\|\boldsymbol{\lambda}\|_{[1]}}}}\right]^{\frac{2+2\nu}{1+3\nu}} \inf_{0 \leqslant \nu \leqslant 1}\left(\frac{M_\nu}{\epsilon}\right)^{\frac{2}{1+3\nu}} \right\rfloor. \tag{17}$$

This worst-case complexity is optimal in the sense of first-order black box models [16].

The line-search procedure at Step 3 of Algorithm 2 also terminates after a finite number of iterations. Similar to Algorithm 1, Algorithm 2 requires 1 gradient query and $i_k$ function evaluations of $g$ at each iteration. The supplementary material proves that the number of oracle queries in Algorithm 2 is upperbounded as follows:

$$N_2(k) \leqslant 2(k+1) + 1 + \frac{1-\nu}{1+\nu}\big[\log_2(k+1) - \log_2(\epsilon)\big] + \frac{2}{1+\nu}\log_2(M_\nu) - \log_2(M_{-1}). \tag{18}$$

Roughly speaking, Algorithm 2 requires approximately two oracle query per iteration on average.

# 5 Numerical experiments

This section illustrates the scalability and the flexibility of our primal-dual framework using some applications in the quantum tomography (QT) and the matrix completion (MC).

## 5.1 Quantum tomography with Pauli operators

We consider the QT problem which aims to extract information from a physical quantum system. A $q$-qubit quantum system is mathematically characterized by its density matrix, which is a complex $p \times p$ positive semidefinite Hermitian matrix $\mathbf{X}^\natural \in \mathcal{S}_+^p$, where $p = 2^q$. Surprisingly, we can provably deduce the state from performing compressive linear measurements $\mathbf{b} = \mathcal{A}(\mathbf{X}) \in \mathcal{C}^n$ based on Pauli operators $\mathcal{A}$ [18]. While the size of the density matrix grows exponentially in $q$, a significantly fewer compressive measurements (i.e., $n = \mathcal{O}(p \log p)$) suffices to recover a pure state $q$-qubit density matrix as a result of the following convex optimization problem:

$$\varphi^\star = \min_{\mathbf{X} \in \mathcal{S}_+^p} \left\{ \varphi(\mathbf{X}) := \frac{1}{2} \|\mathcal{A}(\mathbf{X}) - \mathbf{b}\|_2^2 : \mathrm{tr}(\mathbf{X}) = 1 \right\}, \quad (\mathbf{X}^\star : \varphi(\mathbf{X}^\star) = \varphi^\star), \qquad (19)$$

where the constraint ensures that $\mathbf{X}^\star$ is a density matrix. The recovery is also robust to noise [18].

Since the objective function has Lipschitz gradient and the constraint (i.e., the Spectrahedron) is tuning-free, the QT problem provides an ideal scalability test for both our framework and GCG-type algorithms. To verify the performance of the algorithms with respect to the optimal solution in large-scale, we remain within the noiseless setting. However, the timing and the convergence behavior of the algorithms remain qualitatively the same under polarization and additive Gaussian noise.

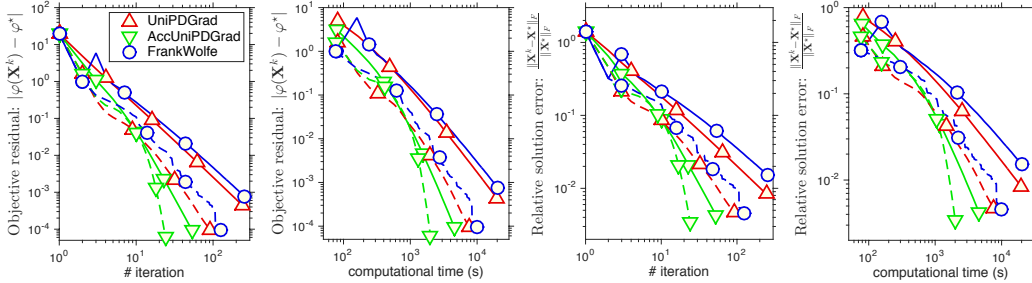

Figure 1: The convergence behavior of algorithms for the $q = 14$ qubits QT problem. The solid lines correspond to the theoretical weighting scheme, and the dashed lines correspond to the line-search (in the weighting step) variants.

To this end, we generate a random pure quantum state (e.g., rank-1 $\mathbf{X}^\natural$), and we take $n = 2p \log p$ random Pauli measurements. For $q = 14$ qubits system, this corresponds to a $268'435'456$ dimensional problem with $n = 317'983$ measurements. We recast (19) into (1) by introducing the slack variable $\mathbf{r} = \mathcal{A}(\mathbf{X}) - \mathbf{b}$.

We compare our algorithms vs. the Frank-Wolfe method, which has optimal convergence rate guarantees for this problem, and its line-search variant. Computing the *sharp*-operator $[\mathbf{x}]^\sharp$ requires a top-eigenvector $\mathbf{e}_1$ of $\mathcal{A}^*(\boldsymbol{\lambda})$, while evaluating $g$ corresponds to just computing the top-eigenvalue $\sigma_1$ of $\mathcal{A}^*(\boldsymbol{\lambda})$ via a power method. All methods use the same subroutine to compute the *sharp*-operator, which is based on MATLAB's `eigs` function. We set $\epsilon = 2 \times 10^{-4}$ for our methods and have a wall-time $2 \times 10^4$s in order to stop the algorithms. However, our algorithms seems insensitive to the choice of $\epsilon$ for the QT problem.

Figure 1 illustrates the iteration and the timing complexities of the algorithms. UniPDGrad algorithm, with an average of $1.978$ line-search steps per iteration, has similar iteration and timing performance as compared to the standard Frank-Wolfe scheme with step-size $\gamma_k = 2/(k+2)$. The line-search variant of Frank-Wolfe improves over the standard one; however, our accelerated variant, with an average of $1.057$ line-search steps, is the clear winner in terms of both iterations and time. We can empirically improve the performance of our algorithms even further by adapting a similar line-search strategy in the weighting step as Frank-Wolfe, i.e., by choosing the weights $w_k$ in a greedy fashion to minimize the objective function. The practical improvements due to line-search appear quite significant.

## 5.2 Matrix completion with MovieLens dataset

To demonstrate the flexibility of our framework, we consider the popular matrix completion (MC) application. In MC, we seek to estimate a low-rank matrix $\mathbf{X} \in \mathbb{R}^{p \times l}$ from its subsampled entries $\mathbf{b} \in \mathbb{R}^n$, where $\mathcal{A}(\cdot)$ is the sampling operator, i.e., $\mathcal{A}(\mathbf{X}) = \mathbf{b}$.

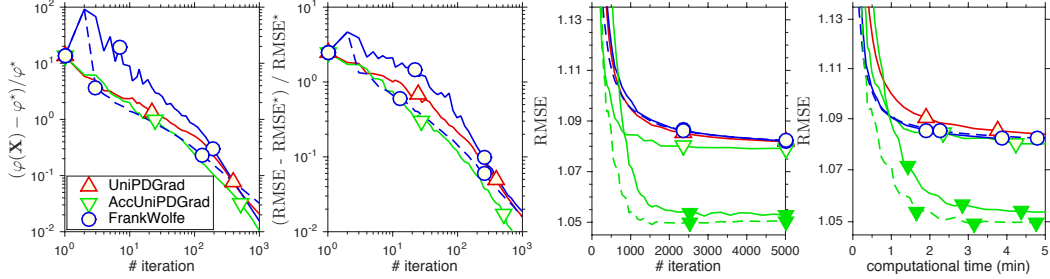

Figure 2: The performance of the algorithms for the MC problems. The dashed lines correspond to the line-search (in the weighting step) variants, and the empty and the filled markers correspond to the formulation (20) and (21), respectively.

Convex formulations involving the nuclear norm have been shown to be quite effective in estimating low-rank matrices from limited number of measurements [19]. For instance, we can solve

$$\min_{\mathbf{X} \in \mathbb{R}^{p \times l}} \left\{ \varphi(\mathbf{X}) = \frac{1}{n} \|\mathcal{A}(\mathbf{X}) - \mathbf{b}\|^2 : \|\mathbf{X}\|_* \leqslant \kappa \right\},\tag{20}$$

with Frank-Wolfe-type methods, where $\kappa$ is a tuning parameter, which may not be available a priori. We can also solve the following parameter-free version

$$\min_{\mathbf{X} \in \mathbb{R}^{p \times l}} \left\{ \psi(\mathbf{X}) = \frac{1}{n} \|\mathbf{X}\|_*^2 : \mathcal{A}(\mathbf{X}) = \mathbf{b} \right\}.\tag{21}$$

While the nonsmooth objective of (21) prevents the tuning parameter, it clearly burdens the computational efficiency of the convex optimization algorithms.

We apply our algorithms to (20) and (21) using the MovieLens 100K dataset. Frank-Wolfe algorithms cannot handle (21) and only solve (20). For this experiment, we did not pre-process the data and took the default `ub` test and training data partition. We start out algorithms form $\boldsymbol{\lambda}_0 = \mathbf{0}^n$, we set the target accuracy $\epsilon = 10^{-3}$, and we choose the tuning parameter $\kappa = 9975/2$ as in [20]. We use `lansvd` function (MATLAB version) from PROPACK [21] to compute the top singular vectors, and a simple implementation of the power method to find the top singular value in the line-search, both with $10^{-5}$ relative error tolerance.

The first two plots in Figure 2 show the performance of the algorithms for (20). Our metrics are the normalized objective residual and the root mean squared error (RMSE) calculated for the test data. Since we do not have access to the optimal solutions, we approximated the optimal values, $\varphi^\star$ and RMSE$^\star$, by 5000 iterations of AccUniPDGrad. Other two plots in Figure 2 compare the performance of the formulations (20) and (21) which are represented by the empty and the filled markers, respectively. Note that, the dashed line for AccUniPDGrad corresponds to the line-search variant, where the weights $w_k$ are chosen to minimize the feasibility gap. Additional details about the numerical experiments can be found in the supplementary material.

## 6  Conclusions

This paper proposes a new primal-dual algorithmic framework that combines the flexibility of proximal primal-dual methods in addressing the general template (1) while leveraging the computational advantages of the GCG-type methods. The algorithmic instances of our framework are *universal* since they can automatically adapt to the unknown Hölder continuity properties implied by the template. Our analysis technique unifies Nesterov's universal gradient methods and GCG-type methods to address the more broadly applicable primal-dual setting. The hallmarks of our approach includes the optimal worst-case complexity and its flexibility to handle nonsmooth objectives and complex constraints, compared to existing primal-dual algorithm as well as GCG-type algorithms, while essentially preserving their low cost iteration complexity.

#### Acknowledgments

This work was supported in part by ERC Future Proof, SNF 200021-146750 and SNF CRSII2-147633. We would like to thank Dr. Stephen Becker of University of Colorado at Boulder for his support in preparing the numerical experiments.

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
