[Supplementary Material]



# A Universal Primal-Dual Convex Optimization Framework

In this supplementary document, we provide the technical proofs and additional implementation details. It is organized as follows: Section A defines the key estimates that form the basis of the universal gradient algorithms. Sections B and C present the proofs of Theorems 4.1 and 4.2 respectively. Finally, Section D provides implementation details of the quantum tomography and the matrix completion problems considered in Section 5.

## A   The key estimate of the proximal-gradient step

Lemma 2 in [1], which we present below as Lemma A.1, provides key properties for constructing universal gradient algorithms. We refer to [1] for the proof of this lemma.

**Lemma A.1.** *Let function $g$ satisfy the Assumption A.2. Then for any $\delta > 0$ and*

$$M \geqslant \left[ \frac{1-\nu}{1+\nu} \frac{1}{\delta} \right]^{\frac{1-\nu}{1+\nu}} M_\nu^{\frac{2}{1+\nu}},$$

*the following statement holds for any $\tilde{\boldsymbol{\lambda}}, \boldsymbol{\lambda} \in \mathbb{R}^n$ :*

$$g(\tilde{\boldsymbol{\lambda}}) \leqslant \underbrace{g(\boldsymbol{\lambda}) + \langle \nabla g(\boldsymbol{\lambda}), \tilde{\boldsymbol{\lambda}} - \boldsymbol{\lambda} \rangle + \frac{M}{2} \|\tilde{\boldsymbol{\lambda}} - \boldsymbol{\lambda}\|^2}_{Q_M(\tilde{\boldsymbol{\lambda}}, \boldsymbol{\lambda})} + \frac{\delta}{2}.$$

This lemma provides an approximate quadratic upper bound for $g$. However, it depends on the choice of the inexactness parameter $\delta$ and the smoothness parameter $\nu$. If $\nu = 1$, then $M$ can be set to the Lipschitz constant $M_1$, and it becomes independent of $\delta$.

The algorithms that we develop in this paper are based on the proximal-gradient step (9) on the dual objective function $G$. This update rule guarantees the following estimate:

**Lemma A.2.** *Let $Q_M$ be the quadratic model of $g$. If $\boldsymbol{\lambda}_{k+1}$, which is defined by (9), satisfies*

$$g(\boldsymbol{\lambda}_{k+1}) \leqslant Q_{M_k}(\boldsymbol{\lambda}_{k+1}; \hat{\boldsymbol{\lambda}}_k) + \frac{\delta_k}{2} \tag{22}$$

*for some $\delta_k \in \mathbb{R}$, then the following inequality holds for any $\boldsymbol{\lambda} \in \mathbb{R}^n$ :*

$$G(\boldsymbol{\lambda}_{k+1}) \leqslant g(\hat{\boldsymbol{\lambda}}_k) + \langle \nabla g(\hat{\boldsymbol{\lambda}}_k), \boldsymbol{\lambda} - \hat{\boldsymbol{\lambda}}_k \rangle + h(\boldsymbol{\lambda}) + \frac{\delta_k}{2} + \frac{M_k}{2} \left[ \|\boldsymbol{\lambda} - \hat{\boldsymbol{\lambda}}_k\|^2 - \|\boldsymbol{\lambda} - \boldsymbol{\lambda}_{k+1}\|^2 \right].$$

*Proof of Lemma A.2.* We note that the optimality condition of (9) is

$$0 \in \nabla g(\hat{\boldsymbol{\lambda}}_k) + M_k(\boldsymbol{\lambda}_{k+1} - \hat{\boldsymbol{\lambda}}_k) + \partial h(\boldsymbol{\lambda}_{k+1}),$$

which can be written as $\hat{\boldsymbol{\lambda}}_k - \boldsymbol{\lambda}_{k+1} \in M_k^{-1}(\nabla g_k(\hat{\boldsymbol{\lambda}}_k) + \partial h(\boldsymbol{\lambda}_{k+1}))$. Let $\nabla h(\hat{\boldsymbol{\lambda}}_{k+1}) \in \partial h(\boldsymbol{\lambda}_{k+1})$ be a subgradient of $h$ at $\boldsymbol{\lambda}_{k+1}$. Then, we have

$$\hat{\boldsymbol{\lambda}}_k - \boldsymbol{\lambda}_{k+1} = \frac{1}{M_k} \left[ \nabla g(\hat{\boldsymbol{\lambda}}_k) + \nabla h(\boldsymbol{\lambda}_{k+1}) \right]. \tag{23}$$

Now, using (23), we can derive

$$\Delta r_k(\boldsymbol{\lambda}) = \frac{1}{2}\|\boldsymbol{\lambda} - \boldsymbol{\lambda}_{k+1}\|^2 - \frac{1}{2}\|\boldsymbol{\lambda} - \hat{\boldsymbol{\lambda}}_k\|^2$$

$$= \langle \hat{\boldsymbol{\lambda}}_k - \boldsymbol{\lambda}_{k+1}, \boldsymbol{\lambda} - \boldsymbol{\lambda}_{k+1}\rangle - \frac{1}{2}\|\hat{\boldsymbol{\lambda}}_k - \boldsymbol{\lambda}_{k+1}\|^2$$

$$\overset{(23)}{=} \frac{1}{M_k}\langle \nabla g(\hat{\boldsymbol{\lambda}}_k) + \nabla h(\boldsymbol{\lambda}_{k+1}), \boldsymbol{\lambda} - \boldsymbol{\lambda}_{k+1}\rangle - \frac{1}{2}\|\hat{\boldsymbol{\lambda}}_k - \boldsymbol{\lambda}_{k+1}\|^2$$

$$= -\frac{1}{M_k}\left[\langle \nabla g(\hat{\boldsymbol{\lambda}}_k), \boldsymbol{\lambda}_{k+1} - \hat{\boldsymbol{\lambda}}_k\rangle + \frac{M_k}{2}\|\boldsymbol{\lambda}_{k+1} - \hat{\boldsymbol{\lambda}}_k\|^2\right]$$

$$\quad + \frac{1}{M_k}\langle \nabla h(\boldsymbol{\lambda}_{k+1}), \boldsymbol{\lambda} - \boldsymbol{\lambda}_{k+1}\rangle + \frac{1}{M_k}\langle \nabla g(\hat{\boldsymbol{\lambda}}_k), \boldsymbol{\lambda} - \hat{\boldsymbol{\lambda}}_k\rangle$$

$$\overset{(22)}{\leqslant} \frac{1}{M_k}\left[g(\hat{\boldsymbol{\lambda}}_k) - g(\boldsymbol{\lambda}_{k+1}) + \frac{\delta_k}{2}\right]$$

$$\quad + \frac{1}{M_k}\langle \nabla h(\boldsymbol{\lambda}_{k+1}), \boldsymbol{\lambda} - \boldsymbol{\lambda}_{k+1}\rangle + \frac{1}{M_k}\langle \nabla g(\hat{\boldsymbol{\lambda}}_k), \boldsymbol{\lambda} - \hat{\boldsymbol{\lambda}}_k\rangle$$

$$\leqslant \frac{1}{M_k}\left[g(\hat{\boldsymbol{\lambda}}_k) - g(\boldsymbol{\lambda}_{k+1}) + \frac{\delta_k}{2}\right]$$

$$\quad + \frac{1}{M_k}\left[h(\boldsymbol{\lambda}) - h(\boldsymbol{\lambda}_{k+1})\right] + \frac{1}{M_k}\langle \nabla g(\hat{\boldsymbol{\lambda}}_k), \boldsymbol{\lambda} - \hat{\boldsymbol{\lambda}}_k\rangle$$

$$= \frac{1}{M_k}\left[g(\hat{\boldsymbol{\lambda}}_k) + \langle \nabla g(\hat{\boldsymbol{\lambda}}_k), \boldsymbol{\lambda} - \hat{\boldsymbol{\lambda}}_k\rangle + h(\boldsymbol{\lambda}) + \frac{\delta_k}{2}\right] - \frac{1}{M_k}G(\boldsymbol{\lambda}_{k+1})$$

where the last inequality directly follows the convexity of $h$. $\qquad\square$

Clearly, (22) holds if $M_k \geqslant \overline{M}_\epsilon$, which is defined by (10), due to Lemma A.1, whenever $\delta_k = \epsilon > 0$.

If $\nu$ and $M_\nu$ are known, we can set $M_k = \overline{M}_\epsilon$, then the condition (22) is automatically satisfied. However, we do not know $\nu$ and $M_\nu$ a priori in general. In this case, $M_k$ can be determined via a line-search procedure on the condition (22).

The following lemma guarantees that the line-search procedure in Algorithms 1 and 2 terminates after a finite number of line-search iterations.

**Lemma A.3.** *The line-search procedure in Algorithm 1 terminates after at most*

$$i_k = \lfloor \log_2(\overline{M}_\epsilon/M_{-1})\rfloor + 1$$

*number of iterations.*

*Similarly, the line-search procedure in Algorithm 2 terminates after at most*

$$i_k = \left\lfloor \log_2\left(\frac{k+1}{\epsilon}\right) + \log_2\left(\frac{M_\nu^{\frac{2}{1+\nu}}}{M_{-1}}\right)\right\rfloor + 1$$

*number of iterations.*

*Proof.* Under Assumption A.2, $M_\nu$ defined in Lemma A.1 is finite. When $\delta_k = \epsilon > 0$ is fixed as in Algorithm 1, the upper bound $\overline{M}_\epsilon = \left[\frac{1-\nu}{(1+\nu)\epsilon}\right]^{\frac{1-\nu}{1+\nu}}M_\nu^{\frac{2}{1+\nu}}$ defined by (10) is also finite. Moreover, the condition (22) holds whenever $M_{k,i} \geqslant \overline{M}_\epsilon$. Since $M_{k,i} = 2M_{k,i-1} = 2^i M_{k,0} \geqslant 2^i M_{-1}$, the linesearch procedure is terminated after at most $i_k = \lfloor \log_2(\overline{M}_\epsilon/M_{-1})\rfloor + 1$ iterations.

Now, we show that the line-search procedure in Algorithm 2 is also finite. By the updating rule of $t_k$, we have $t_{k+1} := 0.5(1 + \sqrt{1 + 4t_k^2}) \leqslant 0.5(1 + (1 + 2t_k)) = t_k + 1$. By induction and $t_0 = 1$, we have $t_k \leqslant k + 1$. Using the definition (10) of $\overline{M}_{\delta_k}$ with $\delta_k = \frac{\epsilon}{t_k}$ and $t_k \leqslant k + 1$, we can show that

$$\overline{M}_{\delta_k} = \left[\frac{1-\nu}{1+\nu}\frac{1}{\delta_k}\right]^{\frac{1-\nu}{1+\nu}}M_\nu^{\frac{2}{1+\nu}} \leqslant \left[\frac{t_k}{\epsilon}\right]^{\frac{1-\nu}{1+\nu}}M_\nu^{\frac{2}{1+\nu}} \leqslant \left[\frac{k+1}{\epsilon}\right]^{\frac{1-\nu}{1+\nu}}M_\nu^{\frac{2}{1+\nu}}. \qquad (24)$$

Next, we note that the condition (22) holds whenever $M_{k,i} \geqslant \overline{M}_{\delta_k}$. However, since $M_{k,i} = 2^i M_{k,0} \geqslant 2^i M_{-1}$, by using (24), it is sufficient to show that the following condition holds for a finite $i$:

$$2^i M_{-1} \geqslant \left[\frac{k+1}{\epsilon}\right]^{\frac{1-\nu}{1+\nu}} M_\nu^{\frac{2}{1+\nu}}.$$

This condition leads to $i \geqslant \log_2\left(\left[\frac{k+1}{\epsilon}\right]^{\frac{1-\nu}{1+\nu}} M_\nu^{\frac{2}{1+\nu}}\right) - \log_2(M_{-1})$. Hence, at the $k$th iteration, we require at most $i_k = \left\lfloor \log_2\left(\frac{k+1}{\epsilon}\right) + \log_2\left(\frac{M_\nu^{\frac{2}{1+\nu}}}{M_{-1}}\right)\right\rfloor + 1$ line-search iterations, which is finite. $\quad\square$

## B Convergence analysis of the universal primal-dual gradient algorithm

In this section, we analyze the convergence of the Algorithm 1 (UniPDGrad). We first provide the convergence guarantee of the dual function in Theorem B.1. Then, we prove the convergence rate and the worst-case complexity given in Theorem 4.1.

### B.1 Convergence rate of the dual objective function

**Theorem B.1.** *Let $\{\boldsymbol{\lambda}_k\}$ be the sequence generated by UniPDGrad. Then,*

$$G(\bar{\boldsymbol{\lambda}}_k) - G(\boldsymbol{\lambda}) \leqslant \bar{G}_k - G(\boldsymbol{\lambda}) \leqslant \frac{\overline{M}_\epsilon}{k+1}\|\boldsymbol{\lambda}_0 - \boldsymbol{\lambda}\|^2 + \frac{\epsilon}{2}, \tag{25}$$

*for any $\boldsymbol{\lambda} \in \mathbb{R}^n$, where $\overline{M}_\epsilon$ is defined by (10) and the two averaging sequences $\{\bar{\boldsymbol{\lambda}}_k\}$ and $\{\bar{G}_k\}$ are defined as follows:*

$$\bar{\boldsymbol{\lambda}}_k := \frac{1}{S_k}\sum_{i=0}^{k}\frac{1}{M_i}\boldsymbol{\lambda}_{i+1} \quad and \quad \bar{G}_k := \frac{1}{S_k}\sum_{i=0}^{k}\frac{1}{M_i}G(\boldsymbol{\lambda}_{i+1}), \quad where \quad S_k := \sum_{i=0}^{k}\frac{1}{M_i}.$$

*Proof.* For $\overline{M}_\epsilon$ defined by (10), since the line-search is successful as shown in Lemma A.1, the condition (22) is satisfied at iteration $i$ with $M_i \leqslant 2\overline{M}_\epsilon$. The following inequality directly follows Lemma A.2 considering the convexity of $g$:

$$G(\boldsymbol{\lambda}_{i+1}) \leqslant G(\boldsymbol{\lambda}) + \frac{\epsilon}{2} + \frac{M_i}{2}\left[\|\boldsymbol{\lambda} - \boldsymbol{\lambda}_i\|^2 - \|\boldsymbol{\lambda} - \boldsymbol{\lambda}_{i+1}\|^2\right], \quad \forall \boldsymbol{\lambda} \in \mathbb{R}^n.$$

Taking the weighted sum of this inequality over $i$, we get

$$\bar{G}_k \leqslant G(\boldsymbol{\lambda}) + \frac{\epsilon}{2} + \frac{1}{2S_k}\left[\|\boldsymbol{\lambda} - \boldsymbol{\lambda}_0\|^2 - \|\boldsymbol{\lambda} - \boldsymbol{\lambda}_{k+1}\|^2\right], \tag{26}$$

for any $\boldsymbol{\lambda} \in \mathbb{R}^n$, and $G(\bar{\boldsymbol{\lambda}}_k) \leqslant \bar{G}_k$ since $G$ is a convex function. Finally, since $M_i \leqslant 2\overline{M}_\epsilon$, we have $S_k \geqslant \frac{(k+1)}{2\overline{M}_\epsilon}$. Substituting this estimate into (26), we obtain (25). $\quad\square$

### B.2 The proof of Theorem 4.1: Convergence rate of the primal sequence

*Proof.* We use the following three expressions to relate the convergence in the dual sequence to the convergence in the primal sequence:

$$\begin{aligned}
g(\boldsymbol{\lambda}_i) &= -f(\mathbf{x}^*(\boldsymbol{\lambda}_i)) + \langle \boldsymbol{\lambda}_i, \mathbf{b} - \mathbf{Ax}^*(\boldsymbol{\lambda}_i)\rangle, \\
\nabla g(\boldsymbol{\lambda}_i) &= \mathbf{b} - \mathbf{Ax}^*(\boldsymbol{\lambda}_i), \\
G(\boldsymbol{\lambda}_{i+1}) &\geqslant G^\star = -d^\star = -f^\star.
\end{aligned} \tag{27}$$

Substituting these expressions into Lemma A.2, we get the following key estimate in the primal space that holds for any $\boldsymbol{\lambda} \in \mathbb{R}^n$:

$$f(\mathbf{x}^*(\boldsymbol{\lambda}_i)) - f^\star \leqslant \langle \mathbf{b} - \mathbf{Ax}^*(\boldsymbol{\lambda}_i), \boldsymbol{\lambda}\rangle + h(\boldsymbol{\lambda}) + \frac{\epsilon}{2} + \frac{M_i}{2}\left[\|\boldsymbol{\lambda} - \boldsymbol{\lambda}_i\|^2 - \|\boldsymbol{\lambda} - \boldsymbol{\lambda}_{i+1}\|^2\right].$$

Taking the weighted sum of this inequality over $i$ and considering the convexity of $f$, we get

$$f(\bar{\mathbf{x}}_k) - f^\star \leqslant \langle \mathbf{b} - \mathbf{A}\bar{\mathbf{x}}_k, \boldsymbol{\lambda} \rangle + h(\boldsymbol{\lambda}) + \frac{\epsilon}{2} + \frac{1}{2S_k} \left[ \|\boldsymbol{\lambda} - \boldsymbol{\lambda}_0\|^2 - \|\boldsymbol{\lambda} - \boldsymbol{\lambda}_{k+1}\|^2 \right]. \tag{28}$$

Setting $\boldsymbol{\lambda} = \mathbf{0}^n$, we get the bound on the right hand side of (15),

$$f(\bar{\mathbf{x}}_k) - f^\star \leqslant \frac{\epsilon}{2} + \frac{\|\boldsymbol{\lambda}_0\|^2}{2S_k} \leqslant \frac{\epsilon}{2} + \frac{\overline{M}_\epsilon \|\boldsymbol{\lambda}_0\|^2}{k+1}.$$

The inequality on the left hand side of (11) follows the following saddle point formulation:

$$f^\star \leqslant \mathcal{L}(\mathbf{x}, \mathbf{r}, \boldsymbol{\lambda}^\star) = f(\mathbf{x}) + \langle \boldsymbol{\lambda}^\star, \mathbf{A}\mathbf{x} - \mathbf{b} - \mathbf{r} \rangle \leqslant f(\mathbf{x}) + \|\boldsymbol{\lambda}^\star\| \|\mathbf{A}\mathbf{x} - \mathbf{b} - \mathbf{r}\|, \tag{29}$$

$\forall \mathbf{r} \in \mathcal{K}$ and $\forall \mathbf{x} \in \mathcal{X}$, where the last inequality holds due to Cauchy-Schwarz inequality. The proof of the convergence rate in the objective residual (11) follows by setting $\mathbf{x} = \bar{\mathbf{x}}$ in (29).

Next, we prove the convergence rate of the feasibility gap (12). We start from the following saddle point formulation:

$$f^\star \leqslant \mathcal{L}(\mathbf{x}, \mathbf{r}, \boldsymbol{\lambda}^\star) = f(\mathbf{x}) + \langle \boldsymbol{\lambda}^\star, \mathbf{A}\mathbf{x} - \mathbf{b} - \mathbf{r} \rangle, \qquad \forall \mathbf{r} \in \mathcal{K}, \ \forall \mathbf{x} \in \mathcal{X}.$$

Substituting this estimate with $\mathbf{x} = \bar{\mathbf{x}}_k$ into (28), we get the following inequality:

$$\langle \mathbf{A}\bar{\mathbf{x}}_k - \mathbf{b} - \mathbf{r}^*(\boldsymbol{\lambda}), \boldsymbol{\lambda} - \boldsymbol{\lambda}^\star \rangle - \frac{1}{2S_k} \left[ \|\boldsymbol{\lambda} - \boldsymbol{\lambda}_0\|^2 - \|\boldsymbol{\lambda} - \boldsymbol{\lambda}_{k+1}\|^2 \right] \leqslant \frac{\epsilon}{2}$$

$$\implies \min_{\mathbf{r} \in \mathcal{K}} \left\{ \langle \mathbf{A}\bar{\mathbf{x}}_k - \mathbf{b} - \mathbf{r}, \boldsymbol{\lambda} - \boldsymbol{\lambda}^\star \rangle - \frac{1}{2S_k} \|\boldsymbol{\lambda} - \boldsymbol{\lambda}_0\|^2 \right\} \leqslant \frac{\epsilon}{2}$$

$$\implies \max_{\boldsymbol{\lambda} \in \mathbb{R}^n} \min_{\mathbf{r} \in \mathcal{K}} \left\{ \langle \mathbf{A}\bar{\mathbf{x}}_k - \mathbf{b} - \mathbf{r}, \boldsymbol{\lambda} - \boldsymbol{\lambda}^\star \rangle - \frac{1}{2S_k} \|\boldsymbol{\lambda} - \boldsymbol{\lambda}_0\|^2 \right\} \leqslant \frac{\epsilon}{2}$$

$$\implies \min_{\mathbf{r} \in \mathcal{K}} \max_{\boldsymbol{\lambda} \in \mathbb{R}^n} \left\{ \langle \mathbf{A}\bar{\mathbf{x}}_k - \mathbf{b} - \mathbf{r}, \boldsymbol{\lambda} - \boldsymbol{\lambda}^\star \rangle - \frac{1}{2S_k} \|\boldsymbol{\lambda} - \boldsymbol{\lambda}_0\|^2 \right\} \leqslant \frac{\epsilon}{2}$$

$$\implies \min_{\mathbf{r} \in \mathcal{K}} \left\{ \langle \mathbf{A}\bar{\mathbf{x}}_k - \mathbf{b} - \mathbf{r}, \boldsymbol{\lambda}_0 - \boldsymbol{\lambda}^\star \rangle + \frac{S_k}{2} \|\mathbf{A}\bar{\mathbf{x}}_k - \mathbf{b} - \mathbf{r}\|^2 \right\} \leqslant \frac{\epsilon}{2}$$

for any $\boldsymbol{\lambda} \in \mathbb{R}^n$, where $\mathbf{r}^*(\boldsymbol{\lambda}) := \arg\max_{\mathbf{r} \in \mathcal{K}} \langle \mathbf{r}, \boldsymbol{\lambda} \rangle$, and the third implication holds due to the Sion's minimax theorem. Hence, there exists a vector $\bar{\mathbf{r}} \in \mathcal{K}$, that satisfies the following inequality:

$$\langle \mathbf{A}\bar{\mathbf{x}}_k - \mathbf{b} - \bar{\mathbf{r}}, \boldsymbol{\lambda}_0 - \boldsymbol{\lambda}^\star \rangle + \frac{S_k}{2} \|\mathbf{A}\bar{\mathbf{x}}_k - \mathbf{b} - \bar{\mathbf{r}}\|^2 \leqslant \frac{\epsilon}{2}.$$

Using Cauchy-Schwarz inequality, this implies

$$-\|\mathbf{A}\bar{\mathbf{x}}_k - \mathbf{b} - \bar{\mathbf{r}}\| \|\boldsymbol{\lambda}_0 - \boldsymbol{\lambda}^\star\| + \frac{S_k}{2} \|\mathbf{A}\bar{\mathbf{x}}_k - \mathbf{b} - \bar{\mathbf{r}}\|^2 \leqslant \frac{\epsilon}{2}.$$

Solving this inequality for $\|\mathbf{A}\bar{\mathbf{x}}_k - \mathbf{b} - \bar{\mathbf{r}}\|$, we get

$$\mathrm{dist}\,(\mathbf{A}\bar{\mathbf{x}}_k - \mathbf{b}, \mathcal{K}) \leqslant \|\mathbf{A}\bar{\mathbf{x}}_k - \mathbf{b} - \bar{\mathbf{r}}\|$$

$$\leqslant \frac{1}{S_k} \left[ \|\boldsymbol{\lambda}_0 - \boldsymbol{\lambda}^\star\| + \sqrt{\|\boldsymbol{\lambda}_0 - \boldsymbol{\lambda}^\star\|^2 + S_k \epsilon} \right]$$

$$\leqslant \frac{1}{S_k} \left[ 2\|\boldsymbol{\lambda}_0 - \boldsymbol{\lambda}^\star\| + \sqrt{S_k \epsilon} \right].$$

We note that $S_k \geqslant \frac{k+1}{2\overline{M}_\epsilon}$, and this completes the proof. $\qquad\square$

### B.2.1 The worst-case complexity analysis

For simplicity, we choose $\boldsymbol{\lambda}_0 = \mathbf{0}^n$ without loss of generality. Then, in order to guarantee both $\mathrm{dist}\,(\mathbf{A}\bar{\mathbf{x}}_k - \mathbf{b}, \mathcal{K}) \leqslant \epsilon$ and $|f(\bar{\mathbf{x}}_k) - f^\star| \leqslant \epsilon$, we require $\left[ \frac{4\overline{M}_\epsilon}{k+1} \|\boldsymbol{\lambda}^\star\| + \sqrt{\frac{2\overline{M}_\epsilon \epsilon}{k+1}} \right] \|\boldsymbol{\lambda}^\star\|_{[1]} \leqslant \epsilon$ due to Theorem 4.1, where $\|\boldsymbol{\lambda}^\star\|_{[1]} := \max\{\|\boldsymbol{\lambda}^\star\|, 1\}$. This leads to (13) as

$$k+1 \geqslant \left[ \frac{4\sqrt{2}\|\boldsymbol{\lambda}^\star\|}{-1 + \sqrt{1 + 8\frac{\|\boldsymbol{\lambda}^\star\|}{\|\boldsymbol{\lambda}^\star\|_{[1]}}}} \right]^2 \frac{\overline{M}_\epsilon}{\epsilon} \implies k_{\max} = \left\lceil \left[ \frac{4\sqrt{2}\|\boldsymbol{\lambda}^\star\|}{-1 + \sqrt{1 + 8\frac{\|\boldsymbol{\lambda}^\star\|}{\|\boldsymbol{\lambda}^\star\|_{[1]}}}} \right]^2 \inf_{0 \leqslant \nu \leqslant 1} \left( \frac{M_\nu}{\epsilon} \right)^{\frac{2}{1+\nu}} \right\rceil.$$

Hence, the worst-case complexity to obtain an $\epsilon$-solution of (1) in the sense of Definition 1.1 is

$$\mathcal{O}\left(\inf_{0\leqslant\nu\leqslant1}\left(\frac{M_\nu}{\epsilon}\right)^{\frac{2}{1+\nu}}\right),$$

which is optimal if $\nu = 0$.

Next, we estimate the total number of oracle quires in UniPDGrad, as in [1]. The total number of oracle quires up to the iteration $k$ is given by $N_1(k) = \sum_{j=0}^{k}(i_j + 1)$. However, since $i_j - 1 = \log_2(M_j/M_{j-1})$, we have

$$N_1(k) = \sum_{j=0}^{k}(i_j + 1) = 2(k+1) + \log_2(M_k) - \log_2(M_{-1}).$$

It remains to use $M_k \leqslant 2\overline{M}_\epsilon$ to obtain (14).

## C   Convergence analysis of the accelerated universal primal-dual algorithm

We now analyze the convergence of AccUniPDGrad (Algorithm 2) in terms of the objective residual and the feasibility gap.

The dual main step of our algorithm is to update $\boldsymbol{\lambda}_{k+1}$ and $\tilde{\boldsymbol{\lambda}}_{k+1}$ from $\hat{\boldsymbol{\lambda}}_k$ and $\tilde{\boldsymbol{\lambda}}_k$ as follows:

$$\begin{cases} \hat{\boldsymbol{\lambda}}_k & := (1-\tau_k)\boldsymbol{\lambda}_k + \tau_k\tilde{\boldsymbol{\lambda}}_k \\ \boldsymbol{\lambda}_{k+1} & := \mathrm{prox}_{M_k^{-1}h}\left(\hat{\boldsymbol{\lambda}}_k - M_k^{-1}\nabla g(\hat{\boldsymbol{\lambda}}_k)\right) \\ \tilde{\boldsymbol{\lambda}}_{k+1} & := \tilde{\boldsymbol{\lambda}}_k - \frac{1}{\tau_k}\left(\hat{\boldsymbol{\lambda}}_k - \boldsymbol{\lambda}_{k+1}\right), \end{cases} \quad (30)$$

where $\tilde{\boldsymbol{\lambda}}_0 = \boldsymbol{\lambda}_0$, $\tau_0 = 1$ and

$$\tau_k^2 = \tau_{k-1}^2(1 - \tau_k). \quad (31)$$

The parameter $M_k$ is determined based on the following line-search condition:

$$g(\boldsymbol{\lambda}_{k+1}) \leqslant g(\hat{\boldsymbol{\lambda}}_k) + \langle\nabla g(\hat{\boldsymbol{\lambda}}_k), \boldsymbol{\lambda}_{k+1} - \hat{\boldsymbol{\lambda}}_k\rangle + \frac{M_k}{2}\|\boldsymbol{\lambda}_{k+1} - \hat{\boldsymbol{\lambda}}_k\|^2 + \frac{\epsilon}{2t_k}, \quad (32)$$

with $M_k \geqslant M_{k-1}$.

Next, we simplify the scheme (30) in the following lemma:

**Lemma C.1.** *The scheme* (30) *can be restated as follows:*

$$\begin{cases} \boldsymbol{\lambda}_{k+1} & := \mathrm{prox}_{M_k^{-1}h}\left(\hat{\boldsymbol{\lambda}}_k - M_k^{-1}\nabla g(\hat{\boldsymbol{\lambda}}_k)\right) \\ t_{k+1} & := \frac{1}{2}\left[1 + \sqrt{1 + 4t_k^2}\right] \\ \hat{\boldsymbol{\lambda}}_{k+1} & := \boldsymbol{\lambda}_{k+1} + \frac{t_k-1}{t_{k+1}}\left(\boldsymbol{\lambda}_{k+1} - \boldsymbol{\lambda}_k\right), \end{cases} \quad (33)$$

*where $\hat{\boldsymbol{\lambda}}_0 = \boldsymbol{\lambda}_0$ and $t_0 = 1$, and $M_k$ is determined based on the line-search condition* (32).

*This dual scheme is of the FISTA form [2], except for the line-search step.*

*Proof.* Let $t_k = \tau_k^{-1}$, then $t_0 = \tau_0^{-1} = 1$. From (30), we have $\tilde{\boldsymbol{\lambda}}_k - \tilde{\boldsymbol{\lambda}}_{k+1} = \frac{1}{\tau_k}(\hat{\boldsymbol{\lambda}}_k - \boldsymbol{\lambda}_{k+1}) = t_k(\hat{\boldsymbol{\lambda}}_k - \boldsymbol{\lambda}_{k+1})$. We also have $\hat{\boldsymbol{\lambda}}_k = (1-\tau_k)\boldsymbol{\lambda}_k + \tau_k\tilde{\boldsymbol{\lambda}}_k$, which leads to $\tilde{\boldsymbol{\lambda}}_k = \frac{1}{\tau_k}[\hat{\boldsymbol{\lambda}}_k - (1-\tau_k)\boldsymbol{\lambda}_k] = t_k[\hat{\boldsymbol{\lambda}}_k - (1 - t_k^{-1})\boldsymbol{\lambda}_k]$. Combining these expressions, we get

$$t_k(\hat{\boldsymbol{\lambda}}_k - \boldsymbol{\lambda}_{k+1}) = \tilde{\boldsymbol{\lambda}}_k - \tilde{\boldsymbol{\lambda}}_{k+1} = t_k[\hat{\boldsymbol{\lambda}}_k - (1 - t_k^{-1})\boldsymbol{\lambda}_k] - t_{k+1}[\hat{\boldsymbol{\lambda}}_{k+1} - (1 - t_{k+1}^{-1})\boldsymbol{\lambda}_{k+1}],$$

and this can be simplified as follows:

$$\begin{aligned} t_{k+1}\hat{\boldsymbol{\lambda}}_{k+1} &= t_k\boldsymbol{\lambda}_{k+1} + t_{k+1}(1 - t_{k+1}^{-1})\boldsymbol{\lambda}_{k+1} - t_k(1 - t_k^{-1})\boldsymbol{\lambda}_k \\ &= (t_k + t_{k+1} - 1)\boldsymbol{\lambda}_{k+1} - (t_k - 1)\boldsymbol{\lambda}_k. \end{aligned}$$

Hence $\hat{\boldsymbol{\lambda}}_{k+1} = \boldsymbol{\lambda}_{k+1} + \frac{t_k-1}{t_{k+1}}(\boldsymbol{\lambda}_{k+1} - \boldsymbol{\lambda}_k)$, which is the third step of (33).

Next, from the condition (31), we have $t_{k+1}^2 - t_{k+1} - t_k^2 = 0$. Hence, $t_{k+1} = \frac{1}{2}\left[1 + \sqrt{1 + 4t_k^2}\right]$, which is exactly the second step of (33). $\qquad\square$

## C.1  The proof of Theorem 4.2: Convergence rate of the primal sequence

*Proof.* From Lemma A.2, we have

$$G(\boldsymbol{\lambda}_{k+1}) \leqslant \big[g(\hat{\boldsymbol{\lambda}}_k) + \langle \nabla g(\hat{\boldsymbol{\lambda}}_k), \boldsymbol{\lambda} - \hat{\boldsymbol{\lambda}}_k \rangle + h(\boldsymbol{\lambda})\big] + \frac{\tau_k \epsilon}{2}$$

$$+ M_k \langle \hat{\boldsymbol{\lambda}}_k - \boldsymbol{\lambda}_{k+1}, \hat{\boldsymbol{\lambda}}_k - \boldsymbol{\lambda} \rangle - \frac{M_k}{2} \|\boldsymbol{\lambda}_{k+1} - \hat{\boldsymbol{\lambda}}_k\|^2 \tag{34}$$

$$\leqslant G(\boldsymbol{\lambda}) + \frac{\tau_k \epsilon}{2} + M_k \langle \hat{\boldsymbol{\lambda}}_k - \boldsymbol{\lambda}_{k+1}, \hat{\boldsymbol{\lambda}}_k - \boldsymbol{\lambda} \rangle - \frac{M_k}{2} \|\boldsymbol{\lambda}_{k+1} - \hat{\boldsymbol{\lambda}}_k\|^2. \tag{35}$$

Note that these inequalities hold $\forall \boldsymbol{\lambda} \in \mathbb{R}^n$. Next, we subtract $G^\star$ from (34) to get

$$G(\boldsymbol{\lambda}_{k+1}) - G^\star \leqslant \big[g(\hat{\boldsymbol{\lambda}}_k) + \langle \nabla g(\hat{\boldsymbol{\lambda}}_k), \boldsymbol{\lambda} - \hat{\boldsymbol{\lambda}}_k \rangle + h(\boldsymbol{\lambda}) - G^\star\big] + \frac{\tau_k \epsilon}{2}$$

$$+ M_k \langle \hat{\boldsymbol{\lambda}}_k - \boldsymbol{\lambda}_{k+1}, \hat{\boldsymbol{\lambda}}_k - \boldsymbol{\lambda} \rangle - \frac{M_k}{2} \|\boldsymbol{\lambda}_{k+1} - \hat{\boldsymbol{\lambda}}_k\|^2, \tag{36}$$

and we set $\boldsymbol{\lambda} = \boldsymbol{\lambda}_k$ in (35), and then subtract $G^\star$ from the both sides, that results in the following inequality:

$$G(\boldsymbol{\lambda}_{k+1}) - G^\star \leqslant G(\boldsymbol{\lambda}_k) - G^\star + \frac{\tau_k \epsilon}{2} - \frac{M_k}{2} \|\boldsymbol{\lambda}_{k+1} - \hat{\boldsymbol{\lambda}}_k\|^2 + M_k \langle \hat{\boldsymbol{\lambda}}_k - \boldsymbol{\lambda}_{k+1}, \hat{\boldsymbol{\lambda}}_k - \boldsymbol{\lambda}_k \rangle. \tag{37}$$

We obtain the following estimate by summing the two inequalities that we get by multiplying (36) by $\tau_k$ and (37) by $(1 - \tau_k)$, and then dividing the resulting estimate by $M_k \tau_k^2$:

$$\frac{1}{M_k \tau_k^2} \big[G(\boldsymbol{\lambda}_{k+1}) - G^\star\big] \leqslant \frac{(1 - \tau_k)}{M_k \tau_k^2} \big[G(\boldsymbol{\lambda}_k) - G^\star\big] + \frac{1}{2}\big[\|\tilde{\boldsymbol{\lambda}}_k - \boldsymbol{\lambda}\|^2 - \|\tilde{\boldsymbol{\lambda}}_{k+1} - \boldsymbol{\lambda}\|^2\big] + \frac{\epsilon}{2 M_k \tau_k}$$

$$+ \frac{1}{M_k \tau_k} \big[g(\hat{\boldsymbol{\lambda}}_k) + \langle \nabla g(\hat{\boldsymbol{\lambda}}_k), \boldsymbol{\lambda} - \hat{\boldsymbol{\lambda}}_k \rangle + h(\boldsymbol{\lambda}) - G^\star\big]. \tag{38}$$

Next, we sum this inequality over $k$ as follows:

$$\sum_{i=0}^{k} \frac{G(\boldsymbol{\lambda}_{i+1}) - G^\star}{M_i \tau_i^2} \leqslant \sum_{i=0}^{k} \Bigg[ \frac{(1 - \tau_i)}{M_i \tau_i^2} \big[G(\boldsymbol{\lambda}_i) - G^\star\big] + \frac{1}{2}\big[\|\tilde{\boldsymbol{\lambda}}_i - \boldsymbol{\lambda}\|^2 - \|\tilde{\boldsymbol{\lambda}}_{i+1} - \boldsymbol{\lambda}\|^2\big] + \frac{\epsilon}{2 M_i \tau_i}$$

$$+ \frac{1}{M_i \tau_i} \big[g(\hat{\boldsymbol{\lambda}}_i) + \langle \nabla g(\hat{\boldsymbol{\lambda}}_i), \boldsymbol{\lambda} - \hat{\boldsymbol{\lambda}}_i \rangle + h(\boldsymbol{\lambda}) - G^\star\big] \Bigg]$$

$$\leqslant \sum_{i=1}^{k} \frac{G(\boldsymbol{\lambda}_i) - G^\star}{M_{i-1} \tau_{i-1}^2} + \frac{1}{2}\big[\|\tilde{\boldsymbol{\lambda}}_0 - \boldsymbol{\lambda}\|^2 - \|\tilde{\boldsymbol{\lambda}}_{k+1} - \boldsymbol{\lambda}\|^2\big] + \frac{\epsilon}{2} \sum_{i=0}^{k} \frac{1}{M_i \tau_i}$$

$$+ \sum_{i=0}^{k} \frac{1}{M_i \tau_i} \big[g(\hat{\boldsymbol{\lambda}}_i) + \langle \nabla g(\hat{\boldsymbol{\lambda}}_i), \boldsymbol{\lambda} - \hat{\boldsymbol{\lambda}}_i \rangle + h(\boldsymbol{\lambda}) - G^\star\big],$$

where the second inequality follows $\tau_0 = 1$ and $\frac{(1 - \tau_k)}{M_k \tau_k^2} \leqslant \frac{1}{M_{k-1} \tau_{k-1}^2}$ for $k = 1, 2, \ldots$, which holds since $M_k \geqslant M_{k-1}$. This implies the followings:

$$0 \leqslant \frac{G(\boldsymbol{\lambda}_{k+1}) - G^\star}{\hat{S}_k M_k \tau_k^2} \leqslant \frac{1}{\hat{S}_k} \sum_{i=0}^{k} \frac{1}{M_i \tau_i} \big[g(\hat{\boldsymbol{\lambda}}_i) + \langle \nabla g(\hat{\boldsymbol{\lambda}}_i), \boldsymbol{\lambda} - \hat{\boldsymbol{\lambda}}_i \rangle + h(\boldsymbol{\lambda}) - G^\star\big]$$

$$+ \frac{1}{2\hat{S}_k} \big[\|\tilde{\boldsymbol{\lambda}}_0 - \boldsymbol{\lambda}\|^2 - \|\tilde{\boldsymbol{\lambda}}_{k+1} - \boldsymbol{\lambda}\|^2\big] + \frac{\epsilon}{2}$$

$$\implies -\frac{1}{\hat{S}_k} \sum_{i=0}^{k} \frac{1}{M_i \tau_i} \big[g(\hat{\boldsymbol{\lambda}}_i) + \langle \nabla g(\hat{\boldsymbol{\lambda}}_i), \boldsymbol{\lambda} - \hat{\boldsymbol{\lambda}}_i \rangle + h(\boldsymbol{\lambda}) - G^\star\big] \leqslant \frac{1}{2\hat{S}_k} \big[\|\tilde{\boldsymbol{\lambda}}_0 - \boldsymbol{\lambda}\|^2 - \|\tilde{\boldsymbol{\lambda}}_{k+1} - \boldsymbol{\lambda}\|^2\big] + \frac{\epsilon}{2}.$$

Now, we use the following expressions to map this estimate into the primal sequence:

$$g(\hat{\boldsymbol{\lambda}}_i) \quad = -f(\mathbf{x}^*(\hat{\boldsymbol{\lambda}}_i)) + \langle \hat{\boldsymbol{\lambda}}_i, \mathbf{b} - \mathbf{A}\mathbf{x}^*(\hat{\boldsymbol{\lambda}}_i) \rangle,$$

$$\nabla g(\hat{\boldsymbol{\lambda}}_i) = \mathbf{b} - \mathbf{A}\mathbf{x}^*(\hat{\boldsymbol{\lambda}}_i),$$

$$G^\star \quad = -d^\star = -f^\star.$$

Then, considering the convexity of $f$, we get

$$f(\bar{\bar{\mathbf{x}}}_k) - f^\star \leqslant \langle \mathbf{b} - \mathbf{A}\bar{\bar{\mathbf{x}}}_k, \boldsymbol{\lambda} \rangle + h(\boldsymbol{\lambda}) + \frac{\epsilon}{2} + \frac{1}{2\hat{S}_k}\left[\|\tilde{\boldsymbol{\lambda}}_0 - \boldsymbol{\lambda}\|^2 - \|\tilde{\boldsymbol{\lambda}}_{k+1} - \boldsymbol{\lambda}\|^2\right]$$

$$\leqslant \frac{\epsilon}{2} + \frac{\|\boldsymbol{\lambda}_0\|^2}{2\hat{S}_k}, \tag{39}$$

where we obtain the second inequality by setting $\boldsymbol{\lambda} = \mathbf{0}^n$.

We can reformulate (31) as $\frac{1}{\tau_k} = \frac{1}{\tau_k^2} - \frac{1}{\tau_{k-1}^2}$. Using this relation, $M_0 \leqslant M_i \leqslant M_k \leqslant 2\overline{M}_{\epsilon\tau_k} = 2t_k^{\frac{1-\nu}{1+\nu}}\overline{M}_\epsilon \leqslant 2(k+2)^{\frac{1-\nu}{1+\nu}}\overline{M}_\epsilon$ and $\frac{k+2}{2} \leqslant t_k < k+2$ for $i = 0, 1, \ldots, k$, we can show that

$$\hat{S}_k := \sum_{i=0}^{k} \frac{1}{M_i \tau_i} \geqslant \sum_{i=0}^{k} \frac{1}{2\overline{M}_{\epsilon\tau_k}\tau_i} = \frac{1}{2\overline{M}_{\epsilon\tau_k}}\left[1 + \sum_{i=1}^{k}\left(\frac{1}{\tau_i^2} - \frac{1}{\tau_{i-1}^2}\right)\right]$$

$$\geqslant \frac{t_k^2}{2(k+2)^{\frac{1-\nu}{1+\nu}}\overline{M}_\epsilon} \geqslant \frac{(k+2)^{\frac{1+3\nu}{1+\nu}}}{8\overline{M}_\epsilon}. \tag{40}$$

We get the bound on the right hand side of (15) by substituting (40) into (39). The inequality on the left hand side of (15) follows the saddle point formulation (29) by setting $\mathbf{x} = \bar{\bar{\mathbf{x}}}_k$.

Finally, we prove the convergence rate in the feasibility gap (16). By the same arguments as in the proof of Theorem 4.1, we have

$$\text{dist}\left(\mathbf{A}\bar{\bar{\mathbf{x}}}_k - \mathbf{b}, \mathcal{K}\right) \leqslant \frac{2\|\boldsymbol{\lambda}_0 - \boldsymbol{\lambda}^\star\|}{\hat{S}_k} + \sqrt{\frac{\epsilon}{\hat{S}_k}}.$$

We complete the proof by substituting (40) into this estimate. $\qquad\square$

## C.2 The worst-case complexity analysis

We analyze the worst-case complexity of AccUniPDGrad algorithm to achieve an $\epsilon$-solution $\bar{\bar{\mathbf{x}}}_k$. For simplicity, we consider the case $\boldsymbol{\lambda}_0 = \mathbf{0}^n$ without loss of generality. Then, we require

$$\left[\frac{16\overline{M}_\epsilon}{(k+2)^{\frac{1+3\nu}{1+\nu}}}\|\boldsymbol{\lambda}^\star\| + \sqrt{\frac{8\overline{M}_\epsilon\epsilon}{(k+2)^{\frac{1+3\nu}{1+\nu}}}}\right]\|\boldsymbol{\lambda}^\star\|_{[1]} \leqslant \epsilon$$

due to the Theorem 4.2, where $\|\boldsymbol{\lambda}^\star\|_{[1]} := \max\{\|\boldsymbol{\lambda}^\star\|, 1\}$. By solving this inequality, we get

$$k + 2 \geqslant \left[\frac{8\sqrt{2}\|\boldsymbol{\lambda}^\star\|}{-1 + \sqrt{1 + 8\frac{\|\boldsymbol{\lambda}^\star\|}{\|\boldsymbol{\lambda}^\star\|_{[1]}}}}\right]^{\frac{2+2\nu}{1+3\nu}}\left[\frac{\overline{M}_\epsilon}{\epsilon}\right]^{\frac{1+\nu}{1+3\nu}}.$$

Using the definition (10) of $\overline{M}_\epsilon$ and considering the fact that $\left[\frac{1-\nu}{1+\nu}\right]^{\frac{1-\nu}{1+\nu}} \leqslant 1$ for $\nu \in [0,1]$, we find the maximum number of iterations that satisfies the above inequality as follows:

$$k_{\max} = \left\lceil\left[\frac{8\sqrt{2}\|\boldsymbol{\lambda}^\star\|}{-1 + \sqrt{1 + 8\frac{\|\boldsymbol{\lambda}^\star\|}{\|\boldsymbol{\lambda}^\star\|_{[1]}}}}\right]^{\frac{2+2\nu}{1+3\nu}}\inf_{0 \leqslant \nu \leqslant 1}\left(\frac{M_\nu}{\epsilon}\right)^{\frac{2}{1+3\nu}}\right\rceil,$$

which is indeed (17).

Hence, the worst-case complexity to obtain an $\epsilon$-solution of (1) in the sense of Definition 1.1 is

$$\mathcal{O}\left(\inf_{0 \leqslant \nu \leqslant 1}\left(\frac{M_\nu}{\epsilon}\right)^{\frac{2}{1+3\nu}}\right),$$

which is optimal in the sense of first-order black box models [3].

Next, we consider the number of oracle quires in AccUniPDGrad. At iteration $k$, the algorithm requires $i_k + 2$ function evaluations of $g$, as we need $i_k + 1$ in the line-search and one for $g(\hat{\boldsymbol{\lambda}}_k)$. Hence, the total number of oracle quires up to the iteration $k$ is $N_2(k) = \sum_{j=0}^{k}(i_j + 2)$. Since $i_j = \log_2(M_j/M_{j-1})$, we have

$$N_2(k) = 2(k+1) + \log_2(M_k) - \log_2(M_{-1}).$$

Using the same argument as in the proof of Lemma A.3, we have $M_k \leqslant 2\overline{M}_{\epsilon\tau_k} \leqslant 2\left[\frac{k+1}{\epsilon}\right]^{\frac{1-\nu}{1+\nu}} M_\nu^{\frac{2}{1+\nu}}$. Hence, we obtain (18) as

$$N_2(k) \leqslant 2(k+1) + 1 + \frac{1-\nu}{1+\nu}\left[\log_2(k+1) - \log_2(\epsilon)\right] + \frac{2}{1+\nu}\log_2(M_\nu) - \log_2(M_{-1}).$$

## D    The implementation details

In this section, we specify key steps of UniPDGrad and AccUniPDGrad for two important applications that we used in Section 5. We also provide an analytic step-size that guarantees the line-search condition without function evaluation.

We performed the experiments in MATLAB, using a computational resource with 4 CPUs of 2.40 GHz and 16 GB memory space for the matrix completion, and 16 CPUs of 2.40 GHz and 512 GB memory space for the quantum tomography problem.

### D.1    Constrained convex optimization involving a quadratic cost

In both quantum tomography and the matrix completion problems, we consider some problem formulations from the following convex optimization template that involves a quadratic cost:

$$\min_{\mathbf{x}\in\mathbb{R}^p}\left\{\frac{1}{2}\|\mathcal{A}(\mathbf{x}) - \mathbf{b}\|^2 : \mathbf{x}\in\mathcal{X}\right\}.$$

For notational simplicity, we consider the problem in $\mathbb{R}^p/\mathbb{R}^n$ spaces in this section, but the ideas apply in general.

Evaluation of the sharp-operator corresponding to the objective function $1/2\|\mathcal{A}(\mathbf{x}) - \mathbf{b}\|^2$ requires a significant computational effort. Yet, by introducing the slack variable $\mathbf{r} = \mathcal{A}(\mathbf{x}) - \mathbf{b}$, we can write an equivalent problem as

$$\min_{(\mathbf{r},\mathbf{x})\in\mathbb{R}^n\times\mathbb{R}^p}\left\{\frac{1}{2}\|\mathbf{r}\|^2 : \mathcal{A}(\mathbf{x}) - \mathbf{r} = \mathbf{b}, \ \mathbf{x}\in\mathcal{X}\right\}.$$

We can write the Lagrange function associated with the linear constraint as

$$\mathcal{L}(\mathbf{r}, \mathbf{x}, \boldsymbol{\lambda}) = \frac{1}{2}\|\mathbf{r}\|^2 + \langle\boldsymbol{\lambda}, \mathbf{r} - \mathcal{A}(\mathbf{x}) + \mathbf{b}\rangle,$$

from which we can derive the (negation of the) dual function

$$g(\boldsymbol{\lambda}) = -\min_{\mathbf{r}\in\mathbb{R}^n, \mathbf{x}\in\mathcal{X}}\mathcal{L}(\mathbf{r}, \mathbf{x}, \boldsymbol{\lambda}) = -\min_{\mathbf{r}\in\mathbb{R}^n}\left\{\frac{1}{2}\|\mathbf{r}\|^2 + \langle\boldsymbol{\lambda}, \mathbf{r}\rangle\right\} + \max_{\mathbf{x}\in\mathcal{X}}\langle\boldsymbol{\lambda}, \mathcal{A}(\mathbf{x})\rangle + \langle\boldsymbol{\lambda}, \mathbf{b}\rangle$$

$$= \frac{1}{2}\|\boldsymbol{\lambda}\|^2 + \langle\boldsymbol{\lambda}, \mathbf{b} - \mathcal{A}(\mathbf{x}^*(\boldsymbol{\lambda}))\rangle, \tag{41}$$

and its subgradient

$$\nabla g(\boldsymbol{\lambda}) = \boldsymbol{\lambda} - \mathbf{b} + \mathcal{A}(\mathbf{x}^*(\boldsymbol{\lambda})),$$

where $\mathbf{x}^*(\boldsymbol{\lambda}) \in [\mathcal{A}^T(\boldsymbol{\lambda})]_{\mathcal{X}}^{\sharp} \equiv \arg\max_{\mathbf{x}\in\mathcal{X}}\langle\mathcal{A}^T(\boldsymbol{\lambda}), \mathbf{x}\rangle$.

For the special case, $\mathcal{X}$ is a norm ball, i.e., $\mathcal{X} \equiv \{\mathbf{x} : \|\mathbf{x}\| \leqslant \kappa\}$, we can simplify (41) as follows:

$$g(\boldsymbol{\lambda}) = \frac{1}{2}\|\boldsymbol{\lambda}\|^2 + \langle\boldsymbol{\lambda}, \mathbf{b}\rangle + \kappa\|\mathcal{A}^T(\boldsymbol{\lambda})\|. \tag{42}$$

**Computing an analytical step-size:** Now, we consider the line-search procedure in UniPDGrad and AccUniPDGrad. Since $h(\boldsymbol{\lambda})$ term is absent in these problems, the line-search condition (22) can be simplified as

$$g(\boldsymbol{\lambda}_{k+1}) = g(\hat{\boldsymbol{\lambda}}_k - \alpha_k \nabla g(\hat{\boldsymbol{\lambda}}_k)) \leqslant g(\hat{\boldsymbol{\lambda}}_k) - \frac{\alpha_k}{2}\|\nabla g(\hat{\boldsymbol{\lambda}}_k)\|^2 + \delta_k/2, \tag{43}$$

where we use the notational convention $\hat{\boldsymbol{\lambda}}_k = \boldsymbol{\lambda}_k$ and $\delta_k = \epsilon$ for UniPDGrad, and $\delta_k = \epsilon/t_k$ for AccUniPDGrad. Using the definition (42), we can upper bound $g(\hat{\boldsymbol{\lambda}}_k - \alpha_k \nabla g(\hat{\boldsymbol{\lambda}}_k))$ by

$$U(\alpha_k) := g(\hat{\boldsymbol{\lambda}}_k) + (\alpha_k^2/2)\|\nabla g(\hat{\boldsymbol{\lambda}}_k)\|^2 - \alpha_k\langle\hat{\boldsymbol{\lambda}}_k - \mathbf{b}, \nabla g(\hat{\boldsymbol{\lambda}}_k)\rangle + \alpha_k\kappa\|\mathcal{A}^T(\nabla g(\hat{\boldsymbol{\lambda}}_k))\|.$$

The condition (43) holds if $U(\alpha_k) = g(\hat{\boldsymbol{\lambda}}_k) - \frac{\alpha_k}{2}\|\nabla g(\hat{\boldsymbol{\lambda}}_k)\|^2 + \delta_k/2$. Solving this second order equation, we obtain $\alpha_k$ explicitly as

$$\alpha_k = \frac{-P + \sqrt{P^2 + 4\delta_k\|\nabla g(\hat{\boldsymbol{\lambda}}_k)\|^2}}{2\|\nabla g(\hat{\boldsymbol{\lambda}}_k)\|^2},$$

where $P := \|\nabla g(\hat{\boldsymbol{\lambda}}_k)\|^2 + 2\kappa\|\mathcal{A}^T(\nabla g(\hat{\boldsymbol{\lambda}}_k))\| - 2\langle\hat{\boldsymbol{\lambda}}_k - \mathbf{b}, \nabla g(\hat{\boldsymbol{\lambda}}_k)\rangle$. Note that, we can use this method to find a good estimate for the initial smoothness constant $M_{-1}$ in the initialization step.

## D.2   Constrained convex optimization involving a norm cost

Now, we consider the second application, which is reformulated as

$$\min_{\mathbf{X}\in\mathbb{R}^{p\times l}}\left\{\psi(\mathbf{X}) = \frac{1}{n}\|\mathbf{X}\|_*^2 : \mathcal{A}(\mathbf{X}) - \mathbf{b} \in \mathcal{K}\right\}.$$

Once again, by introducing the slack variable $\mathbf{r} = \mathcal{A}(\mathbf{X}) - \mathbf{b}$, we get

$$\min_{\mathbf{X}\in\mathbb{R}^{p\times l}, \mathbf{r}\in\mathbb{R}^n}\left\{\frac{1}{n}\|\mathbf{X}\|_*^2 : \mathcal{A}(\mathbf{X}) - \mathbf{r} = \mathbf{b}, \ \mathbf{r} \in \mathcal{K}\right\}.$$

Clearly, the dual components $g$ and $h$ defined in (6) can be expressed as:

$$g(\boldsymbol{\lambda}) = \max_{\mathbf{X}\in\mathbb{R}^{p\times l}}\left\{\langle\mathcal{A}^T(\boldsymbol{\lambda}), \mathbf{X}\rangle - \frac{1}{n}\|\mathbf{X}\|_*^2\right\} + \langle\mathbf{b}, \boldsymbol{\lambda}\rangle = \frac{n}{4}\|\mathcal{A}^T(\boldsymbol{\lambda})\|^2 + \langle\mathbf{b}, \boldsymbol{\lambda}\rangle,$$

$$h(\boldsymbol{\lambda}) = \max_{\mathbf{r}\in\mathcal{K}}\langle-\boldsymbol{\lambda}, \mathbf{r}\rangle = \max_{\|\mathbf{r}\|\leqslant\kappa}\langle-\boldsymbol{\lambda}, \mathbf{r}\rangle = \kappa\|\boldsymbol{\lambda}\|,$$

where $\|\cdot\|$ represents the Euclidean norm for vectors and the spectral norm for matrices. In (21), we consider a special case where $\mathcal{K} \equiv \{\mathbf{0}^n\}$, hence $h(\boldsymbol{\lambda}) = 0$.

Clearly, $\mathbf{X}^*(\boldsymbol{\lambda}) = \frac{n}{2}\sigma_1\mathbf{e}_1\mathbf{e}_1^T \in [\mathcal{A}^T(\boldsymbol{\lambda})]_\psi^\sharp$, where $\sigma_1 = \|\mathcal{A}^T(\boldsymbol{\lambda})\|$ is the top singular value of $\mathcal{A}^T(\boldsymbol{\lambda})$ and $\mathbf{e}_1$ is the associated left singular vector. Hence, we can write the (sub)gradient of g as

$$\nabla g(\boldsymbol{\lambda}) = \mathbf{b} - \mathcal{A}(\mathbf{X}^*(\boldsymbol{\lambda})).$$

We can compute both $\sigma_1$ and $\mathbf{e}_1$ efficiently by using the power method or the Lanczos algorithm.