[Reviews · NeurIPS 2015]

Submitted by Assigned_Reviewer_1

Questions that the authors might want to respond:

1. UPD as I understand from reading [15] is to tackle a special class of functions, that is, when 0 < \nu < 1 which can be thought of as an interpolation of function space between bounded subgradients and lipschitz gradients. But in the paper or in the experiments, the authors do not give any such examples in the machine learning context and hence I'm a bit concerned about the relevance of the proposed algorithms to majority of machine learning problems. Maybe I missed it but can you clearly describe examples of problems in machine learning that have \nu \in (0,1).

2. The algorithms were compared with the vanilla FW and FW with line search for faily well studied problems. This makes it hard for a practitioner to see how this algorithm will perform in reality if one wants to choose between algorithms. For example, let's consider the matrix completion example in section 5.2. For this problem, parallel algorithms are usually preferred like [1] which works very well in practice. It's true that the method proposed here is deterministic and doesn't require a lot of cores as [1] but it will still be useful if the authors can clearly describe the *actual* running time for such problems.

[1] Recht, Benjamin, and Christopher Re. "Parallel stochastic gradient algorithms for large-scale matrix completion." Mathematical Programming Computation 5, no. 2 (2013): 201-226.

3. (Minor) Even though the proofs in the supplementary material is fairly easy to follow, an intuition or an outline of the proof technique will make it easier for the readers.

Comments:

1. Implementation details in section F of the supplement is nice. 2. I liked the comparison between FW and DA subgradient methods in section E.2 in ethe supplement.

Conclusion: The paper proposes methods for a special class of functions and shows experiments that go well with the claim of the authors. Addressing questions 1. and 2. (above) will make me view the work a bit more favorably.
Summary: This paper proposes variants of "Universal" primal dual (UPD) algorithm (due to Nesterov) for machine learning problems. The paper includes a detailed convergence analysis. The submission is easy to read. UPD is adapted for machine learning problems in the sense that the speed of the algorithm isa high priority. Complexity analysis of the proposed algorithms is described clearly. The Accelerated UPD (A-UPD) proposed in section 4.2 performs better than the vanilla UPD in the experiments. For A-UPD, the well known FISTA scheme for composite (smooth + nonsmooth) minimization is adapted to generate the dual iterates which reduces the number of prox computations in each iteration which explains the speed up. Two machine learning problems were used to test the proposed algorithms. The technical proofs in the supplementary file are written in detail. The paper is generally clear some ideas borrowed from well established methods to speed up the algorithms proposed.

Submitted by Assigned_Reviewer_2

Summary:

This paper proposes an optimization framework that uses primal and dual aspects of an optimization problem. It advocates the use of a fenchel operator of the dual function, which is sometimes cheaper to compute than the more commonly used proximal operator. The accelerated algorithm applies to smooth and non-smooth losses, obtaining rates which are optimal (in a sense) in each case; this is called universality in this paper. The experimental results shown are somewhat specialized but strong as far as they extend.

The paper could be clearer about the scope of the contributions: is the Fenchel operator always cheaper? is it significantly so? what are some concrete examples where it is applicable, and some where it is not? It seems more applicable for tasks with matrix hypothesis, are there other applications? For most NIPS readers, deciphering whether their problem fits this algorithms requirements might be an obstacle.

The universality of the title translates to a single algorithm with optimal rates for smooth and non smooth functions; this is an important property, but considering the specialized analysis required to apply the algorithms to a problem (finding and implementing multiple fenchel operators efficiently), the claim is significantly weaker than it would be, for example, for a black box gradient based algorithm. The effect of strong convexity on the efficiency of this algorithm is explored elsewhere; given that for some algorithms the effect is dramatic (sublinear to linear), this is very relevant to the kind of "universality" claimed here.

Minor comments:

- Could be more specific than "Stupendous".

- line 171 "we either have Lipschitz gradient g" -> "either g has Lipschitz gradients"

- "with the our weighting" -> "with our weighting"

- l.307: Say explicitly that the same bound for the same algorithm is simultanuously optimal also for \nu = 0, supporting your title claim; as is it looks as if the unaccelerated version is needed for non-smooth problems.
Summary: A novel and interesting optimization method that covers some smooth and non-smooth problems. The applicability is not as clear as it could be.

Submitted by Assigned_Reviewer_3

The authors presents a framework for the analysis of constrained convex optimization based on Fenchel duality. Rather than the traditionnal proximal operator, this analysis relies on the the sharp operator, a linear minimization oracle, whose computation is generally simpler. The authors formulate a dual problem for which they derive the proximal gradient step as well as a condition on its Hoelder constant. They then introduces two algorithms to solve the dual problem, the second adding a Nesterov-like acceleration scheme. Their methods are said universal as they rely on a line search to automatically adapt to the Holder constant of the dual objective's gradient. The authors present an analysis of the average number of steps involved in this line search. Finally they present several numerical experiments in which they illustrate the advantages of their methods over the Frank-Wolfe algorithm.

The paper seems very sound and rely on different state-of-the-art methods. They provide satisfying results on the computation burden of their line-search step.

The paper is rather clear despite the complexity of its content. The paper might be difficult to read for those unfamiliar with both universal gradients and generalized conditional gradient methods as the authors cover those notions rather briefly, but the authors do cite the appropriate references. Some graphs are hard to read and the captioning could be improved however.

The paper does not propose an entirely novel method as the algorithms combine several existing methods (universal gradients, GCG methods, FISTA acceleration). Those methods however combine rather gracefully into a simple enough algorithmic scheme, and the use of both universal Hoelder inequality and the sharp operaor is far from ubiquitous in the community

The results are not as encouraging as they could be. The unaccelerated method does not seem to perform better than Frank-Wolfe, even with constant step, despite its significant increase in both theoretical and implementation complexity. The accelerated method seems to have a slight edge on Frank-Wolfe with line-search however. Even if in the second setting their algorithm seems more promising, its bad graph formatting makes it hard for the reader to determine the extent to which the accelerated method outperforms Frank-Wolfe. A plot of log((error_min-error)/error_min) where error_min is the RMSE/NMAE for a large number of iterations would really help to assess the performance.

The authors finally presents two settings in which FW-like methods do not apply. As the authors do not compare their results with any baseline it is hard to reach any conclusion, beyond the great acceleration provided by their second scheme. Even naive baselines (like projected sub-gradient) would be useful to assess the quality of the results.

It is however rather satisfying to see an empirical confirmation of the authors' prediction regarding the number of call involved in the line search.

In the end I think the ideas proposed by the authors are quite good, but the paper fail to deliver a convincing case for their use against conditionnal gradient methods yet. A better vizualization of the performance of their algorithms in the experimental part could however convince me otherwise.

l53 dou you mean "may not" rather than cannot? Figure 2: bad formating, hard to read Figure 2 & 3: captioning is lacking, it is not clear to me which is what
Summary: The paper combines gracefully several state-of-the-art methods (universal gradients, GCG methods, FISTA acceleration) into a sound theoretical framework and two rather simple algorithms. The experiments demonstrate a slight edge of the proposed schemes over Frank-Wolfe, but the bad formatting of the graphs and the lack of competing algorithms in the last two experiments prevent the reader from fully appreciating the computational gain. Those drawbacks are easily fixable however.

Submitted by Assigned_Reviewer_4

The paper proposes a primal-dual algorithmic framework for a prototypical constrained convex optimization. The algorithmic framework is designed to adapt to Hoelder continuity properties and to guarantee optimal convergence rates in the objective residual and the feasibility gap. The authors combine Nesterov's universal gradient methods with the generalized conditional gradient-type methods in order to broaden their scope of applicability.

The idea is interesting. Deep coverage of the primal-dual settings and related techniques can be helpful for readers. The manuscript is clearly written and neatly structured, but unkind. Here are some comments.

(1) In each section, the head-paragraph does not state the purpose of the section, which confuses the readers in what to follow.

(2) Has this paper already been published somewhere (including any conference)? If that is the case, the manuscript may not be eligible for NIPS.

Summary: The idea is interesting. Deep coverage of the primal-dual settings and related techniques can be helpful for readers. The manuscript is clearly written and neatly structured, but somewhat unkind.

Author Feedback
Author rebuttal: Reviewer 1.

1. Regrading the Holder continuity with nu in (0,1), as an example, if the objective function f is uniformly convex with an order greater than 2, then its dual function is Holder continuous with nu in (0,1). Hence, loss functions with q-norm greater than 2 would yield nu in (0,1) in the dual. It turns out that such loss functions have theoretical advantages, when the data is quantized (Boufounos et al. "Quantization and Compressed Sensing").

We would like to note however that the boundary cases with the template we provide are still directly relevant to ML. On that front, the accelerated algorithm is rate-optimal across the whole range. Having the same computational platform independent of nu is certainly a practical advantage if a target accuracy is known.

2. We will add the runtimes for the second example as we did for the first example. We will test to see if our method is competitive with [1].

At the same time, we would like to note that the second example is intended to illustrate a desirable flexibility: we can solve non-smooth constrained problems with FW-like iterations (e.g., min_x {||x||_nuc: ||Ax-b||_1 \le kappa}). It is currently not clear how [1] applies to this problem.

Reviewer 2.

1. Fenchel operators are in general cheaper. Please see the discussion in the introduction of Nemirovski and Juditsky [11] (quote: "There are, however, important situations where a cheap LMO indeed is available, but where no proximal setup with easy-to-compute prox-mappings is known.") and the several compelling ML examples based on structured sparsity in Yaoliang Yu's thesis [21].

2. We borrow the term "universal" from Nesterov [15] instead of defining a new concept, and we mean by "universal" that our algorithm enjoys the optimal rates for any Holder degree in the dual, without requiring the Holder parameters.

Strong convexity in general requires a bit of a special treatment in convex algorithms if we want to leverage the property optimally.

Reviewer 3:
1. We agree with the reviewer: We are aware of the formatting of the plots and have already worked on improving the presentation of the figures.

2. The lack of a baseline is related to the fact that to our knowledge, there is no algorithm that addresses the same problem with FW-like iterations. If we did not misunderstood the reviewer, the projected subgradient method might require full SVD's for the problems we consider.

3. The practical gains are quite significant if we actually also use line search in the primal-update for our methods. In fact, for the quantum tomography example, our algorithms exhibit linear convergence with the additional primal line-search steps, which are quite cheap. However, proving the linear convergence requires a special type of analysis, and hence, we use the updates we proved in the main text.

We believe that slight increases in the complexity of the algorithms is warranted given the generality of our approach. However, note that UPD in fact takes less time to reach epsilon than the simpler FW!

Reviewer 4.

In prediction tasks, we do not need to exactly satisfy the data fidelity constraints as in our second example.

Reviewer 6. This paper is not submitted anywhere else.